# Boosting Transferability and Discriminability for Time Series Domain Adaptation

**Mingyang Liu[1], Xinyang Chen[1✉], Yang Shu[2✉], Xiucheng Li[1✉], Weili Guan[3], Liqiang Nie[1]**

[1]School of Computer Science and Technology, Harbin Institute of Technology (Shenzhen)
[2]School of Data Science and Engineering, East China Normal University
[3]School of Electronics and Information Engineering, Harbin Institute of Technology (Shenzhen)
mingyangliu1024@gmail.com, yshu@dase.ecnu.edu.cn
{chenxinyang,lixiucheng,guanweili,nieliqiang}@hit.edu.cn

## Abstract

Unsupervised domain adaptation excels in transferring knowledge from a labeled source domain to an unlabeled target domain, playing a critical role in time series applications. Existing time series domain adaptation methods either ignore frequency features or treat temporal and frequency features equally, which makes it challenging to fully exploit the advantages of both types of features. In this paper, we delve into transferability and discriminability, two crucial properties in transferable representation learning. It's insightful to note that frequency features are more discriminative within a specific domain, while temporal features show better transferability across domains. Based on the findings, we propose **A**dversarial **CO**-learning **N**etworks (**ACON**), to enhance transferable representation learning through a collaborative learning manner in three aspects: (1) Considering the multi-periodicity in time series, multi-period frequency feature learning is proposed to enhance the discriminability of frequency features; (2) Temporal-frequency domain mutual learning is proposed to enhance the discriminability of temporal features in the source domain and improve the transferability of frequency features in the target domain; (3) Domain adversarial learning is conducted in the correlation subspaces of temporal-frequency features instead of original feature spaces to further enhance the transferability of both features. Extensive experiments conducted on a wide range of time series datasets and five common applications demonstrate the state-of-the-art performance of ACON. Code is available at https://github.com/mingyangliu1024/ACON.

## 1 Introduction

Time series classification has achieved significant success in the deep learning era by leveraging discriminative features learned from extensive labeled data [18]. However, the presence of distribution shift may arise when deploying the model, potentially impeding the generalization ability of deep models [31]. Unsupervised domain adaptation [13], offering the potential to transfer knowledge from a labeled source domain to an unlabeled target domain, emerges as a promising solution.

Existing domain adaptation methods tailored for time series primarily focus on learning domain-invariant temporal features [30, 41, 29], yielding promising results. Recently, the significance of frequency features for enhancing domain-invariant representation has also been recognized [16]. However, frequency features and temporal features are treated equally, and their distinct properties are overlooked, leading to the inability to fully leverage both types of features to boost transfer learning.

In this paper, we analyze the two most important properties of features in transfer learning: transferability and discriminability, to investigate the characteristics of the frequency features and temporal

38th Conference on Neural Information Processing Systems (NeurIPS 2024).

features. We find that under the premise of adopting advanced backbones in state-of-the-art works [31, 16], frequency features are more discriminative within a specific domain, while temporal features show better transferability across domains.

Based on the findings, we propose **A**dversarial **CO**-learning **N**etworks (**ACON**) to maximize the potential of temporal features and frequency features in terms both transferability and discriminability in a collaborative learning manner. Firstly, to fully leverage the properties of multi-periodicity in time series, we propose multi-period frequency feature learning to further enhance the discriminability of frequency features. Secondly, we propose temporal-frequency domain mutual learning to enhance the discriminability of temporal features in the source domain and improve the transferability of frequency features in the target domain. Specifically, to harness the potent discriminability of frequency features within the domain, we enable the transfer of knowledge from frequency features to temporal features within the source domain via knowledge distillation. To leverage the strong transferability of temporal features across domains, we facilitate the transfer of knowledge from temporal features to frequency features in the target domain through knowledge distillation. Thirdly, we propose to learn transferable representations via domain adversarial learning in temporal-frequency correlation subspace instead of the original temporal feature space. The temporal-frequency correlation subspace not only possesses the properties of the original temporal feature space and original frequency feature space but also incorporates the correlation between the two types of features. Learning transferable representations in the temporal-frequency correlation subspace can further enhance the transferability of features. Our main contributions can be summarized as follows:

- We uncover the characteristics wherein temporal features and frequency features cannot be equally treated in transfer learning. Specifically, we observe that frequency features are more discriminative within a specific domain, while temporal features show better transferability across domains through empirical findings.

- We design ACON, which enhances UDA in three key aspects: a multi-period feature learning module to enhance the discriminability of frequency features, a temporal-frequency domain mutual learning module to enhance the discriminability of temporal features in the source domain and improve the transferability of frequency features in the target domain, and a domain adversarial learning module in temporal-frequency correlation subspace to further enhance transferability of features.

- Experiments conducted on a wide range of time series datasets and five common applications verify the effectiveness of ACON.

## 2 Related Work

**General Unsupervised Domain Adaptation Methods**   Unsupervised domain adaptation leverages the labeled source domain to predict the labels of a different but related, unlabeled target domain. It finds wide applications in computer vision [46, 15, 8] and natural language processing [40, 39, 44]. Existing UDA methods can be classified into three categories: (1) Methods based on adversarial training aim to learn domain-invariant representations via the game between the feature extractor and the domain discriminator. Widely used methods include DANN [13], CDAN [26] and DIRT-T [34]. (2) Methods based on statistical divergence aim to reduce the domain discrepancy by minimizing domain discrepancy in a latent feature space. Widely used methods include DAN [25], DeepCoral [36] and HoMM [5]. (3) Methods based on self-training produce pseudo-labels on unlabeled data and use confident pseudo-labels together with the labeled data to train the model. Widely used methods include PFAN[6], CST [22] and AdaMatch [2]. However, these methods are generally designed and do not fully leverage the properties of time series. Although these methods can be applied to time series through tailored feature extractors, they often obtain suboptimal performance and UDA algorithm specially designed for time series is needed.

**Unsupervised Domain Adaptation for Time Series**   To date, a few methods have been tailored to unsupervised domain adaptation for time series data. VRADA [30] is the first UDA method for multivariate time series that uses adversarial learning for reducing domain discrepancy. In VRADA, a variational recurrent neural network (VRNN) [10] is trained in an adversarial way to learn domain-invariant temporal features. CoDATS [41] builds upon VRADA but uses a convolutional neural network for the feature extractor, proposing a solution for multi-source domain adaptation in

time series classification. SASA [3] adopts LSTM [33] as feature extractors to capture the domain-invariant association, and aligns sparse associative structure between source and target domain via the minimization of maximum mean discrepancy (MMD) [38]. AdvSKM [23] modifies MMD to make it more suitable for time series data. CLUDA [29] learns contextual representation via contrastive learning, and aligns features between source and target domain via adversarial training. RAINCOAT [16] is the first to introduce frequency features into domain adaptation, aligning temporal features and frequency features respectively via Sinkhorn divergence.

**Research gap** In general, in terms of representation learning, most methods only focus on the temporal domain or assume that the temporal domain and the frequency domain are independent of each other, hindering the full utilization of two types of features. In terms of feature adaptation, existing works only focus on aligning temporal features or adopting simple statistical divergence to align frequency features, ignoring the different properties of the temporal features and frequency features in transfer learning. In terms of evaluation, the existing evaluations are conducted on several datasets of limited scale in a few specific tasks, and more general evaluations are needed.

## 3 Transferability and Discriminability in Time Series

### 3.1 Problem setup

In this paper, we study the UDA problem for time series classification. In time series classification problem, the model receives a set of $n$ labeled samples $\{(\mathbf{x}_i, \mathbf{y}_i)\}_{i=1}^n$, where $i$-th sample $\mathbf{x}_i \in \mathbb{R}^{C \times T}$ contains observation of $C$ variates over $T$ time steps. We allow for both univariate and multivariate time series. In UDA setup, we are given $n_s$ labeled samples from a source domain $\hat{P} = \{(\mathbf{x}_i^s, \mathbf{y}_i^s)\}_{i=1}^{n_s}$ and $n_t$ unlabeled samples from a target domain $\hat{Q} = \{(\mathbf{x}_i^t)\}_{i=1}^{n_t}$, which are sampled from different distributions $P$ and $Q$. Superscripts $s$ and $t$ are adopted to distinguish the source domain and the target domain. UDA for time series aims to learn a time series classification model with labeled source data $\hat{P}$ and unlabeled target data $\hat{Q}$, which can make accurate predictions on the target domain.

In addition to the source domain and target domain in UDA, time series naturally can be represented in the temporal domain and frequency domain. By Fast Fourier Transform (FFT), the raw time series input $\mathbf{x}_i$ in the temporal domain can be transformed to corresponding frequency input $\mathbf{v}_i$ in the frequency domain:

$$\mathbf{v}_i = \text{FFT}(\mathbf{x}_i), \tag{1}$$

where the complex variable $\mathbf{v}_i \in \mathbb{C}^{C \times \lfloor \frac{T}{2} \rfloor}$ contains observation of $C$ variates over $\lfloor \frac{T}{2} \rfloor$ different frequencies. Due to the conjugacy of frequency domain, we only consider the frequencies within $\{1, ..., \lfloor \frac{T}{2} \rfloor\}$.

### 3.2 Discriminability of frequency feature

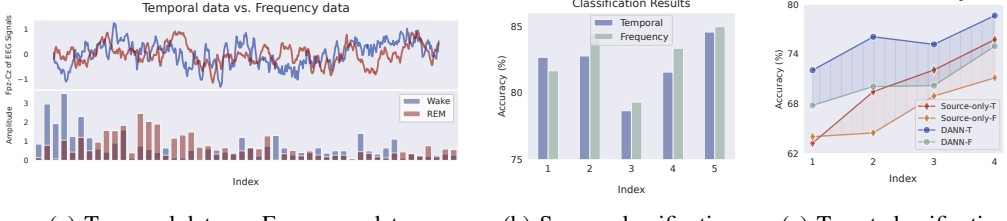

(a) Temporal data vs. Frequency data     (b) Source classification     (c) Target classification

Figure 1: Discriminability of frequency feature: (a) The Electroencephalography (EEG) signal and corresponding frequency data of two classes in the CAP dataset: Wake and Rapid Eye Movement (REM). (b) Classification on the source domain: Temporal domain vs. Frequency domain. (c) Source-only and DANN: Temporal domain vs. Frequency domain.

As Figure 1(a) presented, compared to the uniform distribution of temporal data for different classes, the frequency data for different classes shows distinct differences in the dominant frequencies and peaks, which holds more discriminative information. To further investigate the discriminability of

frequency features, we perform the single data domain classification task in the frequency domain and temporal domain respectively on all five data domains of the CAP [37, 14] dataset.

In order to minimize the impact of specific model structures, we adopt 3-layer 1D-CNN, a generic structure as the temporal feature extractor, and 1-layer linear as the frequency feature extractor, which have both widely validated for their effectiveness in existing time series analysis methods [23, 16, 42, 43]. We only retain the low-frequency data to ensure that the temporal feature extractor and the frequency feature extractor have comparable parameter quantities. For the classifiers, we uniformly use 1-layer linear. As Figure 1(b) shown, with a simple feature extractor, the frequency classification outperforms the temporal classification, demonstrating that the frequency features have better discriminability. More analysis results on different datasets are included in Appendix C.1.

### 3.3 Transferability of temporal feature

Another key criterion that characterizes the performance of domain adaptation is transferability [7]. Transferability indicates the ability to learn invariant features across domains. Since the frequency features have better discriminability within the source domain, it is natural to raise the question: *Will the frequency features also have better discriminability in the target domain?*

We investigate this problem starting with the comparison of four methods: (1) Source-only-F, a model trained in the frequency domain without UDA. (2) Source-only-T, a model trained in the temporal domain without UDA. (3) DANN-F, a model aligning the source features and the target features in the frequency domain via DANN. (4) DANN-T. a model aligning the source features and the target features in the temporal domain via DANN. Figure 1(c) shows the accuracy in the target domains of four source-target domain pairs from the CAP dataset. Compared with Figure 1(b), the frequency classification, which has better discriminability performance in the source domain, actually slightly underperforms in the target domain. It indicates that better discriminability in the source domain does not necessarily imply better discriminability in the target domain. Compared with Source-only methods, the gap between DANN-F and DANN-T is further exacerbated. This suggests that the temporal feature extractor more easily learns domain-invariant features. More analysis results on different datasets are included in Appendix C.2.

The above analysis reveals two insights for time series domain adaptation: With better discriminability but worse transferability, domain adaptation in the frequency domain obtains suboptimal performance; while with better transferability, domain adaptation in the temporal domain has the potential to achieve superior performance under the guidance of more discriminative information.

## 4   Approach

Based on the above observations, our motivation is to simultaneously leverage the strong discriminability of frequency features and the strong transferability of temporal features to enhance domain adaptation. This inspires us to learn domain-invariant temporal and frequency features in a collaborative learning manner.

Figure 2 illustrates the overall structure of our **A**dversarial **CO**-learning **N**etworks (**ACON**). To avoid confusion, subscripts $T$ and $F$ are adopted to distinguish the temporal domain and the frequency domain. Specifically, in the temporal domain, we have a temporal feature extractor with temporal input $\mathbf{f} = \psi_T(\mathbf{x})$ and a temporal classifier $\hat{\mathbf{y}}_T = g_T(\mathbf{f})$; while in the frequency domain, we have a frequency feature extractor with frequency input $\mathbf{z} = \psi_F(\mathbf{v})$ and a frequency classifier $\hat{\mathbf{y}}_F = g_F(\mathbf{z})$. Additionally, we have a domain discriminator $g_D$, which is trained to distinguish the source feature and the target feature. In the following, we will introduce three main contributions in ACON: multi-period frequency feature learning in Section 4.1, temporal-frequency domain mutual learning in Section 4.2, and domain adversarial learning in temporal-frequency correlation subspace in Section 4.3.

### 4.1   Multi-period frequency feature learning

The real-world time series usually present multi-periodicity, which is reflected in the frequency domain as the presence of a few dominant frequencies with significantly larger amplitudes. Data from different periods can have different discriminative patterns. Based on this, before performing FFT, we segment the raw time series according to the top-k significant periods, enhancing the

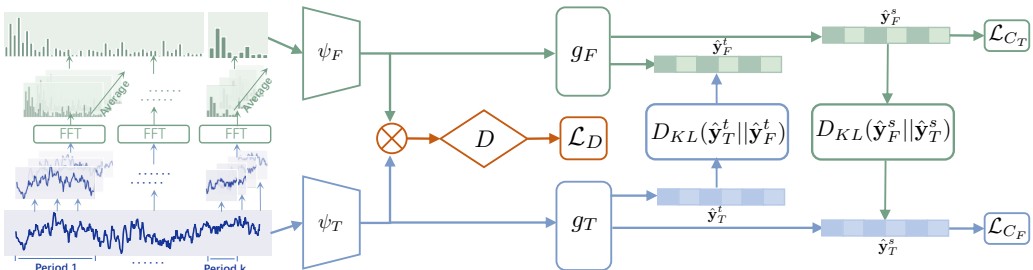

Figure 2: The architecture of ACON. ACON models temporal data (blue) and frequency data (green) simultaneously. Left part: Segment raw frequency data by period to capture different discriminative patterns. Middle part: Align distributions in temporal-frequency correlation subspace via adversarial training. Right part: Mutual learning between the temporal domain and frequency domain.

discriminability of the frequency domain. Additionally, by period-based segmentation, the noises brought by meaningless high frequencies are effectively filtered out [4, 45].

To capture the overall multi-periodicity, before training, we randomly sample mini-batches from the training set to perform FFT and select the frequencies with the top-k amplitudes $\{f_1, \ldots, f_k\}$. Given the frequency $f_j$, the corresponding period is $p_j = \lceil \frac{T}{f_j} \rceil$. For each selected period $p_j$ in $\{p_1, \ldots, p_k\}$ and frequency $f_j$ in the corresponding $\{f_1, \ldots, f_k\}$, we perform the following transform on input $\mathbf{x}_i$:

$$\mathbf{X}_i^j = \text{Reshape}_{p_j}(\mathbf{x}_i), \quad j \in \{1, \ldots, k\},$$
$$\mathbf{v}_i^j = \text{Avg}\left(\text{FFT}\left(\mathbf{X}_i^j\right)\right). \tag{2}$$

where $\mathbf{X}_i^j \in \mathbb{R}^{C \times f_j \times p_j}$, $\mathbf{v}_i^j \in \mathbb{C}^{C \times \lfloor \frac{p_j}{2} \rfloor}$ is averaged from $f_j$ dimensions by $\text{Avg}(\cdot)$. In other words, we perform FFT on each segment obtained by segmenting $\mathbf{x}_i$ with period $p_j$, and average the FFT results across segments to obtain the distribution $\mathbf{v}_i^j$ over the frequencies within $\{1, \ldots, \lfloor \frac{p_j}{2} \rfloor\}$. In this way, we obtain the overall frequency pattern for each period. To keep the discriminative patterns derived from different periods, we concatenate the different $\mathbf{v}_i^j$, obtaining $\mathbf{v}_i$ as the frequency input corresponding to the temporal input $\mathbf{x}_i$:

$$\mathbf{v}_i = \mathbf{v}_i^1 \oplus \ldots \oplus \mathbf{v}_i^k, \quad j \in \{1, \ldots, k\}. \tag{3}$$

We extend the source sample set $\hat{P}$ and the target sample set $\hat{Q}$ to the frequency domain: $\hat{P} = \{(\mathbf{x}_i^s, \mathbf{v}_i^s, \mathbf{y}_i^s)\}_{i=1}^{n_s}$ and $\hat{Q} = \{(\mathbf{x}_i^t, \mathbf{v}_i^t)\}_{i=1}^{n_t}$. To learn features in both real part and imaginary part of complex frequency data, we adopt a complex-valued linear layer as the frequency feature extractor $\psi_F$. Since the phase generally does not provide strong discriminative information, we only retain the amplitudes of each frequency to construct the frequency domain feature $\mathbf{z}_i$:

$$\mathbf{z}_i = \text{Amp}\left(\psi_F(\mathbf{v}_i)\right), \tag{4}$$

where $\text{Amp}(\cdot)$ denotes the calculation of amplitude values. For multivariate time series, we convert $\mathbf{v}_i$ into a single-channel vector by concatenating across different variates.

## 4.2 Temporal-frequency domain mutual learning

Discriminability and transferability are two key criteria that characterize the goodness of feature representations to enable domain adaptation. In Section 3, we reveal that the frequency features are more discriminative within the source domain, while the temporal features are more transferable across domains. Based on this discovery, we propose temporal-frequency domain mutual learning, aiming to leverage the respective advantages of the temporal domain and frequency domain.

The essence of domain mutual learning relies on how to transfer knowledge between the temporal domain and frequency domain. Inspired by model distillation, where the knowledge is transferred by matching the predictions between the teacher and student via the Kullback Leibler (KL) divergence [17], we focus mutual learning on the alignment between the temporal predictions and the frequency predictions. The KL divergence between two predictions $p_1$ and $p_2$ is formulated as:

$$D_{KL}(p_1 || p_2) = \sum_{m=1}^{C} p_1^m \log \frac{p_1^m}{p_2^m}. \tag{5}$$

The KL divergence is asymmetric, that is, $D_{KL}(p_1\|p_2)$ emphasizes aligning $p_2$ to $p_1$, while $D_{KL}(p_2\|p_1)$ emphasizes aligning $p_1$ to $p_2$. Based on the asymmetry, we use different alignment strategies in the source domain and target domain. Specifically, in the source domain, the frequency model serves as a more discriminative teacher, helping the temporal model make more accurate predictions; conversely, in the target domain, the temporal model acts as a more transferable teacher, assisting the frequency model in learning domain-invariant representations. We achieve temporal-frequency domain mutual learning by minimizing the KL Divergence. Formally, domain mutual learning is formulated as:

$$
\begin{aligned}
\mathcal{L}_{M_s}(\psi_T, g_T) &= \mathbb{E}_{(\mathbf{x}_i^s, \mathbf{v}_i^s) \sim \hat{P}}[D_{KL}(\hat{\mathbf{y}}_F^s \| \hat{\mathbf{y}}_T^s)], \\
\mathcal{L}_{M_t}(\psi_F, g_F) &= \mathbb{E}_{(\mathbf{x}_i^t, \mathbf{v}_i^t) \sim \hat{Q}}[D_{KL}(\hat{\mathbf{y}}_T^t \| \hat{\mathbf{y}}_F^t)],
\end{aligned}
\tag{6}
$$

where $\hat{\mathbf{y}}_F^s$ and $\hat{\mathbf{y}}_T^s$ refer to the frequency prediction and temporal prediction in the source domain respectively; while $\hat{\mathbf{y}}_F^t$ and $\hat{\mathbf{y}}_T^t$ refer to the frequency prediction and temporal prediction in the target domain respectively. By aligning $\hat{\mathbf{y}}_T^s$ to $\hat{\mathbf{y}}_F^s$, the training of the temporal feature extractor and classifier is guided with more discriminative information; by aligning $\hat{\mathbf{y}}_F^t$ to $\hat{\mathbf{y}}_T^t$, the transferable knowledge contained in the temporal features is transferred to frequency domain.

## 4.3 Domain adversarial learning in temporal-frequency correlation subspace

Domain adversarial learning [13] is one of the most popular transferable representation learning methods, and it can be employed to learn transferable representation in time series. The key to the effectiveness of the method lies in how to fully utilize two types of features to learn transferable representations. Given time series in temporal domain and frequency domain, domain adversarial learning can be formulated as a minimax optimization problem with three competitive loss terms: (a) $\mathcal{L}_{C_T}$ on the temporal feature extractor $\psi_T$ and classifier $g_T$, which is minimized to guarantee lower source risk of the temporal classifier; (b) $\mathcal{L}_{C_F}$ on the frequency feature extractor $\psi_F$ and classifier $g_F$, which is minimized to guarantee lower source risk of the frequency classifier; (c) $\mathcal{L}_D$ on the temporal feature extractor $\psi_T$, the frequency feature extractor $\psi_F$ and the domain discriminator $g_D$, which is minimized over $g_D$ but maximized over $\psi_T$ and $\psi_F$:

$$
\begin{aligned}
\mathcal{L}_{C_T}(\psi_T, g_T) &= \mathbb{E}_{(\mathbf{x}_i^s, \mathbf{y}_i^s) \sim \hat{P}}[\ell(g_T(\psi_T(\mathbf{x}_i^s)), \mathbf{y}_i^s)], \\
\mathcal{L}_{C_F}(\psi_F, g_F) &= \mathbb{E}_{(\mathbf{v}_i^s, \mathbf{y}_i^s) \sim \hat{P}}[\ell(g_F(\psi_F(\mathbf{v}_i^s)), \mathbf{y}_i^s)], \\
\mathcal{L}_D(\psi_T, \psi_F, g_D) &= -\mathbb{E}_{(\mathbf{x}_i^s, \mathbf{v}_i^s) \sim \hat{P}}\log[g_D(\psi_T(\mathbf{x}_i^s), \psi_F(\mathbf{v}_i^s))] \\
&\quad - \mathbb{E}_{(\mathbf{x}_i^t, \mathbf{v}_i^t) \sim \hat{Q}}\log[1 - g_D(\psi_T(\mathbf{x}_i^t), \psi_F(\mathbf{v}_i^t))],
\end{aligned}
\tag{7}
$$

where $\ell$ denotes cross-entropy loss. Different from standard domain adversarial learning, where there is only one type of feature, domain adversarial learning in time series needs to consider the temporal features and frequency features simultaneously. A simple strategy is to concatenate the temporal feature $\mathbf{f}$ and the frequency feature $\mathbf{z}$. However, with the concatenation strategy, the adversarial game between the domain discriminator and the feature extractors can be viewed as two independent components: the game between $g_D$ and $\psi_T$ and the game between $g_D$ and $\psi_F$. With the worse transferability, $\mathbf{z}$ provides $g_D$ with rich domain-label relevant information. In this case, $g_D$ only needs to focus on the game with $\psi_F$, ignoring the domain adversarial learning in the temporal domain.

To achieve co-alignment in the temporal domain and frequency domain, we propose domain adversarial learning in temporal-frequency correlation subspace. The temporal-frequency correlation subspace not only possesses statistical characteristics of the original temporal feature subspace and original frequency feature subspace but also reflects the correlation between temporal features and frequency features. Reducing the discrepancy of the temporal-frequency correlation subspace not only reduces the discrepancy in the cross-domain temporal and frequency features but also decreases the differences in cross-domain temporal-frequency correlations.

Formally, the vectors in temporal-frequency correlation subspace can be calculated as the outer product $\otimes$ between the temporal feature $\mathbf{f}$ and the frequency feature $\mathbf{z}$:

$$
\mathbf{f} \otimes \mathbf{z} = [\mathbf{z}[1] \cdot \mathbf{f}, \mathbf{z}[2] \cdot \mathbf{f}, \ldots, \mathbf{z}[l] \cdot \mathbf{f}],
\tag{8}
$$

where $\mathbf{z} \in \mathbb{R}^{1 \times l}$. By adjusting the order of dimensions, $\mathbf{z} \otimes \mathbf{f}$ is equivalent to $\mathbf{f} \otimes \mathbf{z}$. Considering the sparsity of the frequency domain and the modeling of long-length time series, the direct outer product

leads to dimension explosion and the sparsity in temporal-frequency feature subspace. We address the problem by performing average pooling over $\mathbf{z}$. Average pooling, which calculates the average value for the amplitudes of neighboring frequencies, yields dense frequency features with smaller dimensions. With outer product and average pooling, the adversarial loss $\mathcal{L}_D$ is formulated as:

$$h(\mathbf{x}_i, \mathbf{v}_i) = \psi_T(\mathbf{x}_i) \otimes \mathrm{P}(\psi_F(\mathbf{v}_i)),$$

$$\mathcal{L}_D(\psi_T, \psi_F, g_D) = -\mathbb{E}_{(\mathbf{x}_i^s, \mathbf{v}_i^s) \sim \hat{P}} \log[g_D(h(\mathbf{x}_i^s, \mathbf{v}_i^s))] - \mathbb{E}_{(\mathbf{x}_i^t, \mathbf{v}_i^t) \sim \hat{Q}} \log[1 - g_D(h(\mathbf{x}_i^t, \mathbf{v}_i^t))], \quad (9)$$

where $h(\cdot)$ denotes the mapping from the inputs of the overall model to the inputs of the domain discriminator, and $\mathrm{P}(\cdot)$ denotes the calculation of average pooling.

### 4.4 Overview

During alignment, our method trains the temporal feature extractor $\psi_T$ and classifier $g_T$ by minimizing the loss $\mathcal{L}_{C_T}$ and trains the frequency feature extractor $\psi_F$ and classifier $g_F$ by minimizing the loss $\mathcal{L}_{C_F}$ using the source sample set $\hat{P}$. Additionally, our method promotes mutual learning between the temporal domain and frequency domain by minimizing the loss $\mathcal{L}_{M_s}$ on the source domain and the loss $\mathcal{L}_{M_t}$ on the target domain. Meanwhile, our method aligns distributions of the source domain and target domain in the temporal-frequency correlation subspace. With two gradient reversal layers between the two feature extractors and the domain discriminator, the adversarial training is achieved by minimizing the loss $\mathcal{L}_D$. To simplify notation, we denote $\theta_F$ as parameters containing $\psi_F$ and $g_F$, and $\theta_T$ is parameters containing $\psi_T$ and $g_T$. The minimax optimization problem is formulated as:

$$\min_{\theta_F, \theta_T} \mathcal{L}_{C_F}(\theta_F) + \mathcal{L}_{C_T}(\theta_T) + \mathcal{L}_{M_s}(\theta_T) + \mathcal{L}_{M_t}(\theta_F) - \mathcal{L}_D(\psi_T, \psi_F, g_D),$$

$$\min_{g_D} \mathcal{L}_D(\psi_T, \psi_F, g_D). \quad (10)$$

## 5 Experiments

### 5.1 Setup

**Datasets**   We conduct extensive experiments using a wide range of time series datasets. (1) Experiments using benchmark datasets in sensor-based human activity recognition (HAR) task: UCIHAR [1], HHAR [35] and WISDM[20]. For HHAR, we first split domains from the perspective of participants, denoted as HHAR-P [16, 31] dataset. Then, we split domains from the perspective of devices, denoted as HHAR-D [12] datasets. (2) Experiments using the benchmark dataset in sleep stage classification (SSC) task: CAP [14, 37]. (3) Experiments using EMG [24, 27] dataset in gesture recognition (GR) task. (4) Experiments using PCL [32, 9, 21, 19] dataset in motor imagery classification (MIC) task. (5) Experiments using FD [31] in machine fault diagnosis (MFD) task. For each dataset, following the existing DA methods on time series [2, 16], we randomly sample 10 source-target domain pairs for evaluation. If the dataset has less than 10 pairs, we evaluate all available domain pairs. Further details, processing and domain splits are included in Appendix A.

**Baselines**   (1) We report the performance of a model without UDA (Source-only) in the temporal domain to show the overall contribution of UDA methods. (2) We implement the following state-of-the-art baselines for UDA of time series data: CODATS[41], AdvSKM[23], CLUDA[29] and RAINCOAT[16]. (3) We additionally implement general unsupervised DA methods: CDAN [26], DeepCoral [36], AdaMatch [2], HoMM [5] and DIRT-T [34].

**Evaluation**   We report accuracy and Macro-F1 Score calculated using target test datasets. Accuracy is computed by dividing the number of correctly classified samples by the total number of samples. Macro-F1 Score is calculated using the unweighted mean of all the per-class F1 scores.

**Implementation**   We adopt the implementation of AdaTime [31] as a benchmarking suite for domain adaptation on time series data, using 1D-CNN as the temporal feature extractor and 1-lyer complex-valued linear as the frequency feature extractor. We use the same feature extractor across all algorithms, ensuring a fair comparison. In all experiments, we use the prediction of the temporal classifier to calculate accuracy and Macro-F1 Score. More experimental details are provided in Appendix B.

Table 1: Average Accuracy (%) on **Eight** Datasets and **Five** Applications for UDA.

| Task | GR | MFD | MI | HAR | | | | SSC |
|------|-----|-----|-----|--------|--------|--------|--------|-----|
| Dataset | EMG | FD | PCL | UCIHAR | HHAR-P | WISDM | HHAR-D | CAP |
| Source-only | 76.24 | 70.04 | 60.95 | 75.12 | 54.25 | 65.78 | 46.82 | 55.86 |
| CDAN | 79.89 | 90.56 | 63.36 | 85.78 | 68.73 | 70.05 | 54.94 | 67.33 |
| DeepCoral | 78.71 | 84.10 | 63.51 | 82.01 | 68.03 | 70.80 | 52.55 | 64.88 |
| AdaMatch | 80.69 | 82.11 | 57.78 | 76.07 | 65.91 | 69.79 | 53.84 | 65.12 |
| HoMM | 78.74 | 85.72 | 63.83 | 80.99 | 65.01 | 67.26 | 52.33 | 65.67 |
| DIRT-T | 79.27 | 88.08 | 61.02 | 83.26 | 64.99 | 69.62 | 56.14 | 70.42 |
| CLUDA | 75.62 | 84.99 | 54.69 | 85.53 | 68.73 | 67.04 | 53.84 | 65.79 |
| AdvSKM | 78.81 | 83.37 | 63.58 | 83.26 | 66.41 | 66.97 | 52.80 | 64.39 |
| CoDATS | 80.60 | 87.20 | 64.18 | 75.54 | 68.71 | 70.66 | 56.27 | 68.23 |
| RAINCOAT | 79.93 | 86.75 | 58.99 | 94.43 | 74.21 | 76.60 | 49.07 | 69.13 |
| **Ours** | **82.91** | **91.74** | **65.02** | **97.02** | **81.74** | **84.80** | **65.04** | **74.08** |
| **Improve(%)** | **2.75** | **1.30** | **1.31** | **2.74** | **10.15** | **10.70** | **15.85** | **5.20** |

## 5.2 Results

Table 1 shows the average accuracy of each method on all datasets and tasks. Overall, our method has won 5 out of 5 tasks and 8 out of 8 datasets (2 metrics). Specifically, our method improves accuracy by 2.75% on GR task, 5.20% on SSC task, 1.31% on MI task, 9.86% on HAR task and 1.30% on MFD task over the advanced baseline on each dataset respectively.

Due to the limited pages, we report the results for selected source-target domain pairs with metric accuracy on the representative datasets EMG (GR task), CAP (SSC task) and HHAR-P (HAR task). More accuracy results are given in Table 11-15. Average macro-f1 score results are given in Table 16. Full macro-f1 score results are given in Table 17-24.

Table 2: Accuracy (%) on CAP for unsupervised domain adaptation.

| Method | 0→1 | 0→3 | 0→4 | 1→0 | 1→4 | 2→3 | 3→0 | 3→1 | 4→1 | 4→3 | Avg |
|--------|-----|-----|-----|-----|-----|-----|-----|-----|-----|-----|-----|
| Source-only | 42.44 | 75.75 | 66.09 | 57.98 | 63.26 | 50.75 | 69.47 | 26.88 | 33.90 | 72.09 | 55.86 |
| CDAN | 68.04 | 77.47 | 72.84 | 62.66 | 67.29 | 53.93 | 72.64 | 63.58 | 58.17 | 76.63 | 67.33 |
| DeepCoral | 67.44 | 77.34 | 72.33 | 59.18 | 67.71 | **58.09** | 70.66 | 53.02 | 49.12 | 73.94 | 64.88 |
| AdaMatch | 60.28 | 77.67 | 75.55 | 68.90 | 63.17 | 37.33 | 73.52 | 59.29 | 60.26 | 75.22 | 65.12 |
| HoMM | 69.89 | 77.11 | 72.27 | 60.36 | 67.89 | 57.96 | 71.58 | 57.61 | 47.52 | 74.49 | 65.67 |
| DIRT-T | 72.16 | 79.21 | 76.04 | 64.18 | 68.69 | 57.75 | 75.47 | 69.91 | 64.59 | 76.16 | 70.42 |
| CLUDA | 67.67 | 75.77 | 58.82 | 70.31 | 70.23 | 53.94 | 74.91 | 53.62 | 58.30 | 74.29 | 65.79 |
| AdvSKM | 63.88 | 77.04 | 72.17 | 60.61 | 66.09 | 58.00 | 70.93 | 55.26 | 46.64 | 73.32 | 64.39 |
| CoDATS | 70.54 | 78.64 | 70.40 | 67.89 | 72.05 | 57.32 | 76.08 | 53.79 | 60.43 | 75.17 | 68.23 |
| RAINCOAT | 70.58 | 72.80 | 73.47 | 65.34 | 69.62 | 56.08 | 71.34 | 70.86 | 70.47 | 70.70 | 69.13 |
| **Ours** | **75.32** | **80.14** | **76.58** | **70.68** | **73.25** | 57.48 | **77.75** | **75.17** | **75.67** | **78.74** | **74.08** |

Table 2 presents the results on CAP dataset. CAP contains over 40,000 samples of 3000 time steps, so adaptation on it is more challenging. Our method outperforms general DA and time series DA methods on 9 out of 10 source-target domain pairs, achieving an average improvement of **5.20%** over the advanced baseline, DIRT-T. Table 3 presents the results on HHAR-P dataset. Our method significantly outperforms RAINCOAT, the state-of-the-art DA method for time series, by **10.15%**.

Table 3: Accuracy (%) on HHAR-P for unsupervised domain adaptation.

| Method | 0→2 | 1→6 | 2→4 | 4→0 | 4→5 | 5→1 | 5→2 | 7→2 | 7→5 | 8→4 | Avg |
|--------|-----|-----|-----|-----|-----|-----|-----|-----|-----|-----|-----|
| Source-only | 64.51 | 70.63 | 45.42 | 32.81 | 78.32 | 90.63 | 25.67 | 32.37 | 39.26 | 62.92 | 54.25 |
| CDAN | 76.19 | 92.57 | 52.57 | 29.09 | 97.27 | 96.16 | 35.04 | 37.05 | 75.26 | 96.11 | 68.73 |
| DeepCoral | 84.23 | 90.14 | 47.08 | 28.13 | 90.49 | 89.91 | 38.39 | 34.45 | 55.73 | 76.88 | 68.03 |
| AdaMatch | 84.78 | 92.31 | 54.50 | 36.45 | 78.45 | 94.20 | 41.96 | 37.65 | 63.80 | 64.69 | 65.91 |
| HoMM | 75.67 | 90.79 | 52.83 | 36.61 | 87.66 | 90.78 | 37.23 | 37.32 | 61.29 | 79.88 | 65.01 |
| DIRT-T | 77.83 | 88.54 | 50.69 | 32.22 | 93.16 | 91.86 | 38.62 | 38.10 | 72.46 | 65.83 | 64.99 |
| CLUDA | 79.84 | 93.40 | 45.90 | 38.84 | 94.08 | 95.57 | 33.93 | 37.80 | 77.57 | 96.52 | 69.35 |
| AdvSKM | 78.94 | 87.91 | 52.57 | 33.49 | 92.64 | 92.71 | 36.53 | 39.95 | 65.49 | 83.75 | 66.41 |
| CoDATS | 79.61 | 90.90 | 60.07 | 21.80 | 97.66 | 97.66 | 41.44 | 38.54 | 58.15 | **97.01** | 68.71 |
| RAINCOAT | **87.72** | 93.33 | 63.75 | 46.46 | **98.05** | 98.25 | 42.63 | 43.32 | 84.17 | 93.75 | 74.21 |
| **Ours** | 86.65 | **93.45** | **79.01** | **53.53** | 97.15 | **98.32** | **65.80** | **65.71** | **88.59** | 89.17 | **81.74** |

Table 4: Ablation studies: Average Accuracy (%) on UCIHAR, HHAR-P and WISDM.

| | $\mathcal{L}_{C_T}$ | $\mathcal{L}_{C_F}$ | Period | $\mathcal{L}_{M_s}$ | $\mathcal{L}_{M_t}$ | $\mathcal{L}_D$ | UCIHAR | HHAR-P | WISDM | Average |
|---|---|---|---|---|---|---|---|---|---|---|
| 1 | ✓ | - | - | - | - | - | 75.12 | 54.25 | 65.78 | 65.05 |
| 2 | - | ✓ | - | - | - | - | 66.88 | 51.08 | 56.47 | 58.14 |
| 3 | - | ✓ | ✓ | - | - | - | 73.47 | 53.16 | 59.10 | 61.91 |
| 4 | ✓ | ✓ | ✓ | - | - | ✓ | 94.05 | 79.49 | 74.19 | 82.58 |
| 5 | ✓ | ✓ | ✓ | ✓ | ✓ | - | 90.83 | 57.83 | 67.77 | 72.14 |
| 6 | ✓ | ✓ | ✓ | ✓ | ✓ | ✓ | **97.02** | **81.74** | **84.80** | **87.85** |

## 5.3 Analysis

**Ablation Study**  We conduct ablation experiments on three datasets, UCIHAR, HHAR-P and WISDM. For each datasets, we select the same 10 source-target domain pairs as mentioned in Section 5.1. The ablation results (average accuracy of 10 domain pairs) are presented in Table 4. We can observe that all learning modules in the proposed method are effective. Further discussions are included in Appendix C.3.

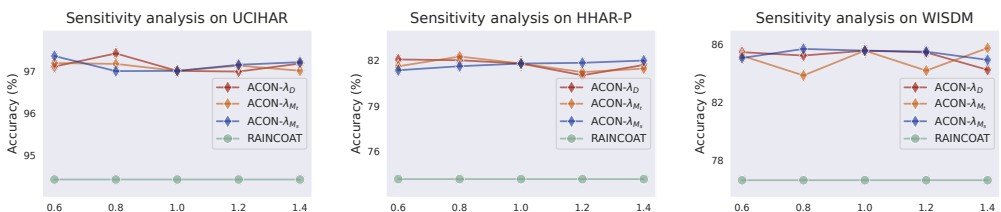

(a) Sensitivity analysis on UCIHAR (b) Sensitivity analysis on HHAR (c) Sensitivity analysis on WISDM

Figure 3: Sensitivity Analysis on three different datasets: (a) UCIHAR (b) HHAR-P (c) WISDM. RAINCOAT: The advanced baseline that achieves suboptimal performance on the three datasets.

**Sensitivity Analysis**  It's worth noting that our total loss in Equation (10) does not include any hyperparameters. In UDA setup, how to search the optimal trade-offs without access to labeled target samples is still an open problem. Considering that, we choose to set all the trade-offs to 1, as it is the most intuitive choice. Without tuning the trade-offs, our proposed ACON still achieves significant improvements. To further investigate the sensitivity of ACON, we update Equation (10) as:

$$
\begin{aligned}
&\min_{\theta_F,\theta_T} \mathcal{L}_{C_F}(\theta_F) + \mathcal{L}_{C_T}(\theta_T) + \lambda_{M_s}\mathcal{L}_{M_s}(\theta_T) + \lambda_{M_t}\mathcal{L}_{M_t}(\theta_F) - \lambda_D\mathcal{L}_D(\psi_T,\psi_F,g_D), \\
&\min_{g_D} \lambda_D\mathcal{L}_D(\psi_T,\psi_F,g_D),
\end{aligned}
\tag{11}
$$

where hyperparameters $\lambda_D$, $\lambda_{M_s}$ and $\lambda_{M_t}$ control the contribution of each component. We investigate the sensitivity of the model to the hyperparameters $\lambda_D$, $\lambda_{M_s}$ and $\lambda_{M_t}$. ACON-$\lambda_D$ refers that the currently investigated hyperparameter is $\lambda_D$, and others are analogous. From Figure 3, we observe that the performance of ACON is quite stable to the hyperparameters in Equation (11). Although setting all the trade-offs to 1 may not achieve the optimal performance, ACON still significantly outperforms the advanced baseline. This implies that ACON can achieve superior performance on a wider range of datasets without the need for careful hyperparameter tuning.

## 6  Conclusion

In this paper, the phenomenon is revealed——that temporal features exhibit better transferability across domains, whereas frequency features tend to be more discriminative within a specific domain. Based on the findings, **A**dversarial **CO**-learning **N**etworks (**ACON**) is proposed to boost the transferability and discriminability in a collaborative learning manner. Specifically, multi-period feature learning is proposed to enhance the discriminability of frequency features; temporal-frequency domain mutual learning is proposed to enhance the discriminability of temporal features in the source domain and improve the transferability of frequency features in the target domain; domain adversarial learning in temporal-frequency correlation subspace is proposed to further enhance transferability of features. ACON achieves state-of-the-art performance on a wide range of time series datasets.

## Acknowledgements

This work was supported by the National Natural Science Foundation of China (62306085, 62206074, 62406112, 62476071, 62236003), Shenzhen College Stability Support Plan (GXWD20231130151329002, GXWD20220811173233001, GXWD20220817144428005).

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

## A  Dataset

### A.1  Detailed Statistics

We conduct extensive experiments using a wide range of time series datasets. The detailed statistics for each dataset is included in Table 5. For EMG dataset, we use the processed version released by DIVERSIFY [28]. For PCL, CAP and HHAR-D datasets, we use the processed versions released by WOODS [11]. For UCIHAR, HHAR-P, WISDM and FD datasets, we use the processed versions released by AdaTime [31].

Table 5: Summary of datasets.

| Dataset | Subjects | Channels | Length | Class | Total | Task |
|---------|----------|----------|--------|-------|-------|------|
| EMG | 4 | 8 | 200 | 6 | 6883 | GR |
| FD | 4 | 1 | 5120 | 3 | 10916 | FD |
| PCL | 3 | 48 | 750 | 2 | 22598 | MIC |
| CAP | 5 | 19 | 3000 | 6 | 40387 | SSC |
| UCIHAR | 30 | 9 | 128 | 6 | 3290 | HAR |
| HHAR-P | 9 | 3 | 128 | 6 | 17934 | HAR |
| WISDM | 30 | 3 | 128 | 6 | 2070 | HAR |
| HHAR-D | 5 | 6 | 500 | 6 | 13674 | HAR |

### A.2  Dataset Processing

Each domain of datasets is randomly divided into 80% training, and 20% testing. We follow [31], apply Z-score normalization to both the training and testing splits of the data, using the following equation:

$$x_i^{normalize} = \frac{x_i - x^{mean}}{x^{std}}, \quad i = 1, 2, \ldots, N \tag{12}$$

where $N = N_s$ for the source domain data and $N = N_t$ for the target domain data. Note that both the training and testing splits are normalized based on the training set statistics only.

## B  Experimental Details

The experiments were conducted on a single NVIDIA GeForce696 RTX 4090 with 24GiB of memory. As shown in Figure 3, without tuning the trade-offs of training loss, ACON still achieves significant improvements. Here we report other key hyperparameters for ACON in Table 6. Additional hyperparameters can be found in our code. In all experiments, we adopt 3-layer 1D-CNN as the temporal feature extractor (the specific structure is kept consistent with the existing works [31, 16]). For frequency feature extraction, we adopt a 1-layer complex-valued linear as the frequency feature extractor.

Table 6: Key hyperparameters for ACON.

| Hyperparameter | EMG | FD | PCL | CAP | UCIHAR | HHAR-P | WISDM | HHAR-D |
|----------------|-----|-----|-----|-----|--------|--------|-------|--------|
| Epoch | 50 | 50 | 50 | 50 | 50 | 50 | 50 | 50 |
| Batch Size | 32 | 32 | 32 | 32 | 32 | 32 | 32 | 32 |
| Learning Rate | 0.001 | 0.01 | 0.001 | 0.001 | 0.01 | 0.001 | 0.003 | 0.01 |

## C  Further Analysis

### C.1  Discriminability of Frequency Feature

In Table 7, we report the average accuracy of classification experiments under the setting of Section 3.2 using five different datasets: UCIAHR[1], HHAR-P[35], WISDM[20], CAP[14, 37] and FD[31]. For each dataset, we collect all the domains involved in the selected 10 domain pairs as mentioned in Section 5.1, and perform the classification task on them.

Table 7: Classification Accuracy (%) in the source domain: Temporal domain vs. Frequency domain.

| Dataset | UCIHAR | HHAR-P | WISDM | CAP | FD |
|---|---|---|---|---|---|
| Temporal domain | 86.18 | 97.01 | 95.63 | 81.63 | 97.99 |
| Frequency domain | 95.69 | 97.77 | 98.01 | 82.73 | 98.83 |

## C.2 Transferability of Temporal Feature

In Table 8, we report the average accuracy of classification experiments under the setting of Section 3.3 using three different datasets: UCIAHR[1], HHAR-P[35], WISDM[20], CAP[14, 37] and FD[31]. For each dataset, we perform the domain adaptation and classification task on the selected 10 domain pairs as mentioned in Section 5.1.

Table 8: Classification Accuracy (%) in the target domain: Temporal domain vs. Frequency domain.

| Dataset | UCIHAR | HHAR-P | WISDM | CAP | FD |
|---|---|---|---|---|---|
| Source-only-T | 75.12 | 54.25 | 65.78 | 70.14 | 70.04 |
| Source-only-F | 66.88 | 51.08 | 56.47 | 67.15 | 69.53 |
| DANN-T | 88.30 | 72.57 | 71.54 | 75.49 | 86.21 |
| DANN-F | 80.64 | 68.73 | 60.99 | 70.76 | 81.46 |

## C.3 Ablation Study

### C.3.1 Ablation Study on different modules

We conduct ablation experiments on three datasets, UCIHAR, HHAR-P and WISDM. For each datasets, we select the same 10 source-target domain pairs as mentioned in Section 5.1. The ablation results (average accuracy of 10 domain pairs) are presented in Table 9. We verify the effectiveness of all learning modules in the proposed method by answering the following questions.

***Can multi-period frequency feature learning enhance the discriminability of frequency feature?*** By comparing the 2nd and 3rd rows, we can observe that with multi-period frequency feature learning, the model makes more accurate predictions on the target domain. Meanwhile, compared with 1st row, even with multi-period frequency feature learning, the performance on the target domain is still inferior to the classification on the temporal domain, which is consistent with our conclusion in Section 3.3 that the temporal features have better transferability.

***Can aligning distributions in temporal-frequency subspace effectively learn domain-invariant features?*** By comparing the 1st, 3rd and 4th rows, we can observe that distribution alignment significantly improves the performance. It indicates that by aligning the distributions in the temporal-frequency subspace, the model learns more domain-invariant features.

***Can temporal-frequency domain mutual learning leverage the respective advantages?*** By comparing the 1st, 3rd and 5th rows, we can observe that the model with domain mutual learning outperforms the model using only the temporal domain or only the frequency domain. It demonstrates that via domain mutual learning, the temporal domain and frequency domain successfully transfer meaningful knowledge, leveraging their respective advantages.

***Can the different modules mutually promote each other?*** By comparing the 3rd, 4th, 5th and 6th rows, we can observe that with all modules, the model achieves the optimal performance. Specifically, with multi-period frequency feature learning, the frequency domain transfers more discriminative knowledge to the temporal domain; with aligning distribution in temporal-frequency subspace, the temporal domain transfers more transferable knowledge to the frequency domain; with aligning distribution in temporal-frequency subspace and the temporal domain as a more tranferbale teacher, both the temporal domain and frequency domain learn domain-invariant features; with domain mutual learning, the frequency domain and the temporal domain enhance each other in the transfer progress, achieving a synergistic effect.

Table 9: Ablation study on different modules: Average Accuracy (%) on UCIHAR, HHAR-P and WISDM.

| | $\mathcal{L}_{C_T}$ | $\mathcal{L}_{C_F}$ | Period | $\mathcal{L}_{M_s}$ | $\mathcal{L}_{M_t}$ | $\mathcal{L}_D$ | UCIHAR | HHAR-P | WISDM | Average |
|---|---|---|---|---|---|---|---|---|---|---|
| 1 | ✓ | - | - | - | - | - | 75.12 | 54.25 | 65.78 | 65.05 |
| 2 | - | ✓ | - | - | - | - | 66.88 | 51.08 | 56.47 | 58.14 |
| 3 | - | ✓ | ✓ | - | - | - | 73.47 | 53.16 | 59.10 | 61.91 |
| 4 | ✓ | ✓ | ✓ | - | - | ✓ | 94.05 | 79.49 | 74.19 | 82.58 |
| 5 | ✓ | ✓ | ✓ | ✓ | ✓ | - | 90.83 | 57.83 | 67.77 | 72.14 |
| 6 | ✓ | ✓ | ✓ | ✓ | ✓ | ✓ | **97.02** | **81.74** | **84.80** | **87.85** |

# D  Limitations

Although ACON boosts transferability and discriminability for time series Domain adaptation, like existing DA methods, it is still unstable enough in time series with relatively large variances. This is a problem that needs to be solved urgently in the future.

# E  Broader Impacts

We investigate how to boost Transferability and discriminability for domain adaptation in time series classification. We reveal that the frequency features are more discriminative, while the temporal features are more transferable. Upon this, we propose multi-period frequency feature learning, domain mutual learning, and distribution alignment in temporal-frequency feature subspace. The purpose of our research is to advance the research progress in the relevant community without any negative social impact.

# F  Full Resluts

We present all experimental results in this section. Notably, our model achieves superior performance, yielding improvements of more than 6% in terms of accuracy and 4% in terms of Macro-F1 across 8 cross-domain time series datasets and 5 common applications on average.

Table 10: Accuracy (%) on EMG for unsupervised domain adaptation.

| Method | 0→1 | 0→2 | 0→3 | 1→2 | 1→3 | 2→0 | 2→1 | 2→3 | 3→1 | 3→2 | Avg |
|---|---|---|---|---|---|---|---|---|---|---|---|
| Source-only | 84.94 | 74.38 | 73.38 | 74.38 | 73.88 | 73.88 | 82.16 | 73.69 | 79.38 | 72.31 | 76.24 |
| CDAN | 87.84 | 76.63 | 77.63 | 77.44 | 81.63 | 73.94 | 87.10 | 75.13 | 83.98 | 77.63 | 79.89 |
| DeepCoral | 87.50 | 76.44 | 76.19 | 77.63 | 77.63 | 74.69 | 84.72 | 75.50 | 81.93 | 74.88 | 78.71 |
| AdaMatch | 89.03 | 75.94 | 79.38 | 76.94 | 80.00 | 76.31 | 89.94 | 81.31 | 84.26 | 73.81 | 80.69 |
| HoMM | 87.61 | 76.50 | 75.75 | 77.00 | 77.94 | 73.94 | 84.89 | 75.88 | 82.61 | 75.31 | 78.74 |
| DIRT-T | 89.77 | 75.25 | 78.69 | 75.88 | 80.06 | 70.63 | 84.77 | 77.69 | 83.30 | 76.69 | 79.27 |
| CLUDA | 78.18 | 75.00 | 76.75 | 74.75 | 74.19 | 75.94 | 79.43 | 70.00 | 76.88 | 75.13 | 75.62 |
| AdvSKM | 86.42 | 75.94 | 76.25 | 77.25 | 78.00 | 74.88 | 85.06 | 77.25 | 81.76 | 75.31 | 78.81 |
| CoDATS | 88.24 | 77.44 | 78.31 | 78.44 | 81.81 | 73.75 | 86.65 | 78.88 | 84.43 | 78.06 | 80.60 |
| RAINCOAT | 89.60 | 77.00 | 78.56 | 78.25 | 83.13 | 73.06 | 85.68 | 76.88 | 83.13 | 74.00 | 79.93 |
| **Ours** | **92.50** | **79.06** | **81.75** | **80.13** | **83.13** | **77.94** | **90.91** | **79.75** | **85.11** | **78.88** | **82.91** |

Table 11: Accuracy (%) on PCL for unsupervised domain adaptation.

| Method | 0→1 | 0→2 | 1→0 | 1→1 | 2→0 | 2→1 | Avg |
|---|---|---|---|---|---|---|---|
| Source-only | 65.68 | 57.79 | 59.65 | 60.65 | 56.51 | 65.46 | 60.95 |
| CDAN | 68.37 | 59.77 | 62.79 | 62.44 | 57.50 | 69.27 | 63.36 |
| DeepCoral | 67.66 | 60.08 | 62.59 | 63.58 | 57.76 | 69.36 | 63.51 |
| AdaMatch | 66.11 | 54.92 | 58.30 | 58.73 | 53.75 | 54.88 | 57.78 |
| HoMM | 68.28 | 60.27 | 62.80 | 63.75 | 58.71 | 69.17 | 63.83 |
| DIRT-T | 61.69 | 57.29 | 60.77 | 62.13 | 56.79 | 67.47 | 61.02 |
| CLUDA | 56.60 | 53.46 | 60.13 | 57.79 | 49.78 | 50.35 | 54.69 |
| AdvSKM | 67.62 | 59.90 | 63.06 | 64.15 | 58.07 | 68.68 | 63.58 |
| CoDATS | 70.52 | 57.83 | **65.10** | 64.17 | 57.83 | 69.62 | 64.18 |
| RAINCOAT | 58.46 | 54.04 | 59.88 | 60.81 | 57.63 | 63.14 | 58.99 |
| **Ours** | **70.63** | **60.58** | 63.42 | **64.63** | **60.32** | **70.53** | **65.02** |

Table 12: Accuracy (%) on FD for unsupervised domain adaptation.

| Method | 0→1 | 0→2 | 0→3 | 1→0 | 1→2 | 2→0 | 2→1 | 2→3 | 3→0 | 3→2 | Avg |
|---|---|---|---|---|---|---|---|---|---|---|---|
| Source-only | 62.21 | 53.71 | 62.41 | 63.91 | 73.95 | 64.08 | 93.17 | 95.54 | 57.08 | 74.31 | 70.04 |
| CDAN | **91.29** | 71.83 | **90.13** | 96.50 | 90.09 | 83.10 | 99.38 | 99.98 | 95.40 | 87.95 | 90.56 |
| DeepCoral | 75.54 | 71.79 | 76.03 | 89.13 | 83.55 | 76.34 | 98.84 | 98.55 | 87.50 | 83.71 | 84.10 |
| AdaMatch | 67.81 | 55.38 | 62.88 | 92.21 | 98.57 | 79.08 | 89.96 | 90.40 | 87.23 | 97.57 | 82.11 |
| HoMM | 81.54 | 71.63 | 78.17 | 89.89 | 84.78 | 76.03 | 98.71 | 99.55 | 90.94 | 85.96 | 85.72 |
| DIRT-T | 75.94 | 70.85 | 76.36 | 98.10 | 90.27 | 81.92 | **100.0** | 99.99 | 97.06 | 90.29 | 88.08 |
| CLUDA | 90.47 | **82.63** | 88.68 | 89.06 | 92.23 | 61.92 | 93.91 | 90.80 | 82.01 | 78.17 | 84.99 |
| AdvSKM | 74.71 | 66.05 | 73.30 | 87.86 | 86.29 | 76.85 | 98.66 | 99.38 | 84.89 | 85.74 | 83.37 |
| CoDATS | 81.79 | 73.26 | 83.15 | 89.22 | 88.68 | 81.43 | 99.89 | **100.0** | 85.47 | 89.00 | 87.20 |
| RAINCOAT | 85.18 | 79.40 | 89.04 | 78.84 | 90.11 | 81.43 | 95.18 | 96.81 | 77.39 | 94.08 | 86.75 |
| **Ours** | 86.52 | 69.00 | 86.96 | 97.92 | **99.80** | **84.29** | 98.62 | 98.93 | **97.72** | **97.66** | **91.74** |

Table 13: Accuracy (%) on UCIHAR for unsupervised domain adaptation.

| Method | 2→11 | 6→23 | 7→13 | 9→18 | 12→16 | 13→19 | 18→21 | 20→6 | 23→13 | 24→12 | Avg |
|---|---|---|---|---|---|---|---|---|---|---|---|
| Source-only | 76.56 | 67.36 | 83.68 | 24.65 | 61.11 | 88.89 | 100.0 | 94.10 | 71.18 | 83.68 | 75.12 |
| CDAN | 85.42 | 87.50 | 92.01 | 58.86 | 66.67 | 96.52 | 100.0 | 95.13 | 82.64 | 93.40 | 85.78 |
| DeepCoral | 90.63 | 84.38 | 87.50 | 46.88 | 65.28 | 95.49 | 100.0 | 95.49 | 69.79 | 87.50 | 82.01 |
| AdaMatch | 75.00 | 80.20 | 85.76 | 56.59 | 49.65 | 94.79 | 100.0 | 84.37 | 68.75 | 70.83 | 76.07 |
| HoMM | 74.06 | 82.71 | 81.88 | 73.96 | 70.21 | 96.67 | 98.75 | 73.33 | 77.71 | 80.63 | 80.99 |
| DIRT-T | 80.21 | 74.31 | 82.99 | 59.03 | 67.01 | 99.30 | 98.61 | 92.36 | 74.72 | 94.27 | 83.26 |
| CLUDA | 81.77 | 92.01 | 99.31 | 67.71 | 65.28 | 94.44 | 98.96 | 97.22 | 72.92 | 99.31 | 85.53 |
| AdvSKM | 98.96 | 88.54 | 92.71 | 74.65 | 69.44 | 93.05 | 100.0 | 85.41 | 79.51 | 96.87 | 83.26 |
| CoDATS | 68.23 | 74.31 | 77.43 | 63.89 | 66.32 | 94.09 | 99.65 | 70.49 | 56.25 | 82.81 | 75.54 |
| RAINCOAT | 100.0 | 95.83 | **100.0** | 75.69 | **86.52** | 100.0 | 100.0 | 93.41 | 86.52 | 93.75 | 94.43 |
| **Ours** | **100.0** | **96.25** | 99.16 | **91.66** | 85.63 | **100.0** | **100.0** | **97.50** | **100.0** | **100.0** | **97.02** |

Table 14: Accuracy (%) on WISDM for unsupervised domain adaptation.

| Method | 2→32 | 4→15 | 7→30 | 12→7 | 12→19 | 18→20 | 20→30 | 21→31 | 25→29 | 26→2 | Avg |
|---|---|---|---|---|---|---|---|---|---|---|---|
| Source-only | 81.16 | 79.86 | 89.32 | 71.53 | 54.29 | 83.74 | 67.96 | 21.29 | 26.11 | 82.52 | 65.78 |
| CDAN | 89.37 | 65.97 | 84.79 | 70.48 | 51.01 | 88.62 | 77.02 | 46.58 | 44.33 | 83.33 | 70.05 |
| DeepCoral | 87.92 | 62.50 | 91.26 | 79.86 | 51.77 | 64.23 | 81.88 | 54.62 | 53.89 | 77.44 | 70.80 |
| AdaMatch | 74.39 | 78.47 | 89.64 | 73.26 | 55.30 | 75.20 | 74.76 | 31.32 | 57.78 | 87.20 | 69.79 |
| HoMM | 77.10 | 74.58 | 78.64 | 68.13 | 50.61 | 71.22 | 72.82 | 56.39 | 57.00 | 66.10 | 67.26 |
| DIRT-T | 77.78 | 70.83 | 90.61 | 70.20 | 51.51 | 85.36 | 71.84 | 54.41 | 60.04 | 66.46 | 69.62 |
| CLUDA | 73.91 | 67.36 | 86.40 | 65.97 | 49.24 | 83.74 | 72.49 | 49.97 | 35.00 | 86.47 | 67.04 |
| AdvSKM | 70.83 | 95.85 | 93.85 | 77.08 | 47.47 | 81.30 | 21.28 | 44.45 | 74.79 | 74.95 | 66.97 |
| CoDATS | 77.29 | 70.83 | 83.20 | 70.17 | 47.47 | 76.01 | 82.85 | 52.61 | 53.89 | 83.29 | 70.66 |
| RAINCOAT | 79.71 | **97.91** | 91.28 | 89.80 | **85.00** | 92.23 | 91.66 | 59.09 | 82.97 | 83.50 | 76.60 |
| **Ours** | **89.86** | 86.25 | **98.06** | **98.13** | 77.73 | 83.66 | 91.26 | **63.61** | 60.00 | **99.51** | **84.80** |

Table 15: Accuracy (%) on HHAR-D for unsupervised domain adaptation.

| Method | 0→1 | 0→2 | 0→3 | 0→4 | 1→0 | 1→3 | 1→4 | 2→1 | 3→4 | 4→1 | Avg |
|---|---|---|---|---|---|---|---|---|---|---|---|
| Source-only | 65.48 | 33.59 | 31.71 | 39.79 | 34.69 | 44.83 | 49.54 | 38.17 | 86.17 | 44.23 | 46.82 |
| CDAN | 69.86 | 48.28 | 38.22 | 48.42 | 48.75 | 60.48 | 51.33 | 47.84 | 87.33 | 48.89 | 54.94 |
| DeepCoral | 68.94 | 42.88 | 40.67 | 47.96 | 35.63 | 55.31 | 56.21 | 44.71 | 87.25 | 45.96 | 52.55 |
| AdaMatch | 71.78 | 39.60 | 39.74 | 47.50 | 52.50 | 55.48 | 58.33 | 46.49 | 85.83 | 41.15 | 53.84 |
| HoMM | 69.66 | 40.51 | 39.16 | 50.42 | 35.94 | 55.02 | 57.13 | 42.36 | 86.79 | 46.35 | 52.33 |
| DIRT-T | 68.37 | 42.14 | 47.21 | 52.92 | 41.25 | 60.14 | 55.63 | 46.73 | 92.25 | **54.81** | 56.14 |
| CLUDA | 71.78 | 39.60 | 39.74 | 47.50 | 52.50 | 55.48 | 58.33 | 46.49 | 85.83 | 41.15 | 53.84 |
| AdvSKM | 67.93 | 40.71 | 40.19 | 47.33 | 37.19 | 55.65 | 59.54 | 42.69 | 87.46 | 49.33 | 52.80 |
| CoDATS | 72.50 | 43.35 | 50.79 | 45.50 | 58.44 | 62.24 | 54.54 | 40.14 | 89.63 | 45.53 | 56.27 |
| RAINCOAT | 74.47 | 36.52 | 48.82 | 35.29 | 51.25 | 41.49 | 41.50 | 34.28 | 88.58 | 38.46 | 49.07 |
| **Ours** | **77.50** | **61.36** | **54.69** | **65.46** | **69.38** | **71.30** | **62.13** | **50.10** | **93.63** | 44.86 | **65.04** |

Table 16: Average Macro-F1 Score on Eight Datasets and Five Applications for UDA.

| Task | GR | MFD | MI | HAR | | | | SSC |
|---|---|---|---|---|---|---|---|---|
| Dataset | EMG | FD | PCL | UCIHAR | HHAR-P | WISDM | HHAR-D | CAP |
| Source-only | 0.76 | 0.65 | 0.60 | 0.73 | 0.50 | 0.52 | 0.43 | 0.52 |
| CDAN | 0.80 | 0.92 | 0.63 | 0.86 | 0.68 | 0.54 | 0.53 | 0.62 |
| DeepCoral | 0.79 | 0.81 | 0.63 | 0.82 | 0.62 | 0.52 | 0.49 | 0.59 |
| AdaMatch | 0.81 | 0.78 | 0.56 | 0.76 | 0.62 | 0.54 | 0.51 | 0.57 |
| HoMM | 0.79 | 0.81 | 0.64 | 0.79 | 0.64 | 0.49 | 0.49 | 0.60 |
| DIRT-T | 0.79 | 0.88 | 0.61 | 0.81 | 0.64 | 0.54 | 0.53 | 0.64 |
| CLUDA | 0.75 | 0.82 | 0.48 | 0.86 | 0.67 | 0.57 | 0.51 | 0.59 |
| AdvSKM | 0.79 | 0.80 | 0.63 | 0.87 | 0.65 | 0.55 | 0.49 | 0.59 |
| CoDATS | 0.81 | 0.88 | 0.63 | 0.72 | 0.63 | 0.56 | 0.55 | 0.62 |
| RAINCOAT | 0.80 | 0.89 | 0.59 | 0.93 | 0.75 | 0.74 | 0.47 | 0.59 |
| **Ours** | **0.83** | **0.93** | **0.65** | **0.97** | **0.80** | **0.74** | **0.62** | **0.67** |

Table 17: Macro-F1 Score on EMG for unsupervised domain adaptation.

| Method | 0→1 | 0→2 | 0→3 | 1→2 | 1→3 | 2→0 | 2→1 | 2→3 | 3→1 | 3→2 | Avg |
|---|---|---|---|---|---|---|---|---|---|---|---|
| Source-only | 0.85 | 0.74 | 0.74 | 0.74 | 0.75 | 0.75 | 0.82 | 0.74 | 0.78 | 0.72 | 0.76 |
| CDAN | 0.88 | 0.77 | 0.78 | 0.78 | 0.82 | 0.74 | 0.87 | 0.76 | 0.84 | 0.78 | 0.80 |
| DeepCoral | 0.87 | 0.76 | 0.76 | 0.78 | 0.78 | 0.75 | 0.84 | 0.76 | 0.82 | 0.75 | 0.79 |
| AdaMatch | 0.89 | 0.76 | 0.79 | 0.77 | 0.80 | 0.76 | 0.90 | 0.81 | 0.84 | 0.74 | 0.81 |
| HoMM | 0.87 | 0.77 | 0.76 | 0.77 | 0.78 | 0.74 | 0.84 | 0.76 | 0.82 | 0.75 | 0.79 |
| DIRT-T | 0.90 | 0.75 | 0.79 | 0.76 | 0.80 | 0.71 | 0.84 | 0.78 | 0.83 | 0.77 | 0.79 |
| CLUDA | 0.78 | 0.75 | 0.77 | 0.75 | 0.74 | 0.76 | 0.79 | 0.70 | 0.75 | 0.75 | 0.75 |
| AdvSKM | 0.86 | 0.76 | 0.76 | 0.77 | 0.78 | 0.76 | 0.85 | 0.77 | 0.81 | 0.75 | 0.79 |
| CoDATS | 0.88 | 0.77 | 0.78 | 0.79 | 0.82 | 0.74 | 0.86 | 0.79 | 0.84 | 0.78 | 0.81 |
| RAINCOAT | 0.89 | 0.77 | 0.79 | 0.78 | 0.83 | 0.73 | 0.85 | 0.77 | 0.83 | 0.74 | 0.80 |
| **Ours** | **0.92** | **0.79** | **0.82** | **0.80** | **0.83** | **0.78** | **0.91** | **0.80** | **0.85** | **0.79** | **0.83** |

Table 18: Macro-F1 Score on CAP for unsupervised domain adaptation.

| Method | 0→1 | 0→3 | 0→4 | 1→0 | 1→4 | 2→3 | 3→0 | 3→1 | 4→1 | 4→3 | Avg |
|---|---|---|---|---|---|---|---|---|---|---|---|
| Source-only | 0.39 | 0.71 | 0.61 | 0.50 | 0.54 | 0.45 | 0.63 | 0.30 | 0.33 | 0.69 | 0.52 |
| CDAN | 0.62 | 0.73 | 0.66 | 0.58 | 0.59 | 0.48 | 0.68 | 0.61 | 0.55 | 0.73 | 0.62 |
| DeepCoral | 0.61 | 0.73 | 0.65 | 0.54 | 0.58 | 0.53 | 0.66 | 0.44 | 0.44 | 0.70 | 0.59 |
| AdaMatch | 0.52 | 0.73 | 0.64 | 0.58 | 0.55 | 0.29 | 0.66 | 0.52 | 0.51 | 0.67 | 0.57 |
| HoMM | 0.62 | 0.73 | 0.65 | 0.56 | 0.59 | 0.54 | 0.66 | 0.50 | 0.48 | 0.71 | 0.60 |
| DIRT-T | **0.65** | 0.75 | **0.69** | 0.57 | 0.59 | 0.50 | 0.69 | 0.67 | 0.59 | 0.71 | 0.64 |
| CLUDA | 0.58 | 0.71 | 0.51 | 0.63 | 0.61 | 0.44 | 0.67 | 0.50 | 0.55 | 0.68 | 0.59 |
| AdvSKM | 0.58 | 0.73 | 0.65 | 0.55 | 0.59 | 0.53 | 0.66 | 0.48 | 0.41 | 0.69 | 0.59 |
| CoDATS | 0.64 | 0.75 | 0.65 | 0.61 | 0.63 | 0.51 | 0.70 | 0.51 | 0.54 | 0.71 | 0.62 |
| RAINCOAT | 0.58 | 0.65 | 0.61 | 0.55 | 0.56 | 0.50 | 0.62 | 0.63 | 0.60 | 0.61 | 0.59 |
| **Ours** | 0.64 | **0.76** | 0.68 | **0.62** | **0.63** | **0.54** | **0.71** | **0.72** | **0.70** | **0.74** | **0.67** |

Table 19: Macro-F1 Score on PCL for unsupervised domain adaptation.

| Method | 0→1 | 0→2 | 1→0 | 1→1 | 2→0 | 2→1 | Avg |
|---|---|---|---|---|---|---|---|
| Source-only | 0.65 | 0.57 | 0.58 | 0.60 | 0.55 | 0.64 | 0.60 |
| CDAN | 0.68 | 0.59 | 0.62 | 0.62 | 0.57 | 0.69 | 0.63 |
| DeepCoral | 0.68 | 0.60 | 0.62 | 0.63 | 0.57 | 0.69 | 0.63 |
| AdaMatch | 0.66 | 0.53 | 0.58 | 0.57 | 0.53 | 0.51 | 0.56 |
| HoMM | 0.68 | 0.60 | 0.63 | 0.63 | 0.58 | 0.69 | 0.64 |
| DIRT-T | 0.61 | 0.57 | 0.61 | 0.62 | 0.56 | 0.67 | 0.61 |
| CLUDA | 0.55 | 0.49 | 0.59 | 0.56 | 0.33 | 0.36 | 0.48 |
| AdvSKM | 0.67 | 0.60 | 0.63 | 0.63 | 0.58 | 0.69 | 0.63 |
| CoDATS | 0.70 | 0.55 | **0.65** | 0.64 | 0.57 | 0.69 | 0.63 |
| RAINCOAT | 0.58 | 0.54 | 0.59 | 0.61 | 0.57 | 0.63 | 0.59 |
| **Ours** | **0.71** | **0.60** | 0.63 | **0.64** | **0.60** | **0.71** | **0.65** |

Table 20: Macro-F1 Score on FD for unsupervised domain adaptation.

| Method | 0→1 | 0→2 | 0→3 | 1→0 | 1→2 | 2→0 | 2→1 | 2→3 | 3→0 | 3→2 | Avg |
|---|---|---|---|---|---|---|---|---|---|---|---|
| Source-only | 0.41 | 0.33 | 0.41 | 0.65 | 0.77 | 0.64 | 0.95 | 0.97 | 0.59 | 0.78 | 0.65 |
| CDAN | **0.91** | 0.76 | 0.90 | 0.95 | 0.92 | 0.86 | 1.00 | 1.00 | 0.94 | 0.91 | 0.92 |
| DeepCoral | 0.61 | 0.62 | 0.62 | 0.90 | 0.87 | 0.77 | 0.99 | 0.99 | 0.89 | 0.88 | 0.81 |
| AdaMatch | 0.50 | 0.45 | 0.46 | 0.91 | 0.98 | 0.80 | 0.93 | 0.93 | 0.87 | 0.97 | 0.78 |
| HoMM | 0.61 | 0.52 | 0.62 | 0.91 | 0.88 | 0.78 | 0.99 | 1.00 | 0.91 | 0.89 | 0.81 |
| DIRT-T | 0.80 | 0.62 | 0.70 | **0.97** | 0.93 | 0.84 | **1.00** | 1.00 | 0.96 | 0.93 | 0.88 |
| CLUDA | 0.84 | 0.80 | 0.79 | 0.88 | 0.93 | 0.50 | 0.95 | 0.90 | 0.84 | 0.80 | 0.82 |
| AdvSKM | 0.55 | 0.54 | 0.57 | 0.89 | 0.89 | 0.76 | 0.99 | **1.00** | 0.87 | 0.89 | 0.80 |
| CoDATS | 0.80 | 0.69 | 0.87 | 0.90 | 0.92 | 0.86 | 1.00 | 1.00 | 0.87 | 0.92 | 0.88 |
| RAINCOAT | 0.89 | **0.84** | **0.92** | 0.81 | 0.92 | 0.85 | 0.96 | 0.98 | 0.81 | 0.94 | 0.89 |
| **Ours** | 0.86 | 0.75 | 0.89 | 0.96 | **1.00** | **0.88** | 0.99 | 0.99 | **0.96** | **0.98** | **0.93** |

Table 21: Macro-F1 Score on UCIHAR for unsupervised domain adaptation.

| Method | 2→11 | 6→23 | 7→13 | 9→18 | 12→16 | 13→19 | 18→21 | 20→6 | 23→13 | 24→12 | Avg |
|---|---|---|---|---|---|---|---|---|---|---|---|
| Source-only | 0.69 | 0.63 | 0.84 | 0.17 | 0.58 | 0.91 | 1.00 | 0.94 | 0.71 | 0.84 | 0.73 |
| CDAN | 0.85 | 0.88 | 0.91 | 0.61 | 0.64 | 0.97 | 1.00 | 0.95 | 0.82 | 0.92 | 0.86 |
| DeepCoral | 0.91 | 0.81 | 0.87 | 0.44 | 0.65 | 0.95 | 1.00 | 0.95 | 0.70 | 0.88 | 0.82 |
| AdaMatch | 0.73 | 0.81 | 0.86 | 0.55 | 0.48 | 0.94 | 1.00 | 0.84 | 0.67 | 0.70 | 0.76 |
| HoMM | 0.73 | 0.78 | 0.81 | 0.69 | 0.69 | 0.96 | 0.99 | 0.71 | 0.75 | 0.78 | 0.79 |
| DIRT-T | 0.81 | 0.68 | 0.82 | 0.58 | 0.62 | 0.99 | 0.98 | 0.92 | 0.74 | 0.93 | 0.81 |
| CLUDA | 0.81 | 0.92 | 0.99 | 0.67 | 0.64 | 0.94 | 0.99 | 0.98 | 0.71 | 0.99 | 0.86 |
| AdvSKM | 0.99 | 0.87 | 0.92 | 0.73 | 0.68 | 0.93 | 1.00 | 0.84 | 0.77 | 0.96 | 0.87 |
| CoDATS | 0.66 | 0.71 | 0.78 | 0.60 | 0.64 | 0.93 | 0.99 | 0.65 | 0.54 | 0.81 | 0.72 |
| RAINCOAT | 1.00 | 0.96 | **1.00** | 0.76 | 0.86 | 1.00 | 1.00 | 0.94 | 0.86 | 0.94 | 0.93 |
| **Ours** | **1.00** | **0.97** | 0.99 | **0.91** | **0.86** | **1.00** | **1.00** | **0.98** | **1.00** | **1.00** | **0.97** |

Table 22: Macro-F1 Score on HHAR-P for unsupervised domain adaptation.

| Method | 0→2 | 1→6 | 2→4 | 4→0 | 4→5 | 5→1 | 5→2 | 7→2 | 7→5 | 8→4 | Avg |
|---|---|---|---|---|---|---|---|---|---|---|---|
| Source-only | 0.60 | 0.64 | 0.32 | 0.29 | 0.78 | 0.90 | 0.19 | 0.31 | 0.36 | 0.58 | 0.50 |
| CDAN | 0.70 | 0.93 | 0.52 | 0.27 | **0.98** | 0.98 | 0.35 | 0.32 | 0.76 | **0.97** | 0.68 |
| DeepCoral | 0.86 | 0.91 | 0.45 | 0.26 | 0.90 | 0.90 | 0.36 | 0.32 | 0.50 | 0.73 | 0.62 |
| AdaMatch | 0.83 | 0.93 | 0.46 | 0.32 | 0.76 | 0.94 | 0.40 | 0.37 | 0.60 | 0.61 | 0.62 |
| HoMM | 0.70 | 0.91 | 0.45 | 0.37 | 0.88 | 0.91 | 0.34 | 0.40 | 0.61 | 0.79 | 0.64 |
| DIRT-T | 0.76 | 0.86 | 0.51 | 0.30 | 0.93 | 0.90 | 0.36 | 0.34 | 0.73 | 0.64 | 0.64 |
| CLUDA | 0.82 | **0.94** | 0.44 | 0.40 | 0.94 | 0.96 | 0.37 | 0.36 | 0.65 | 0.84 | 0.67 |
| AdvSKM | 0.72 | 0.88 | 0.44 | 0.33 | 0.93 | 0.92 | 0.35 | 0.41 | 0.64 | 0.83 | 0.65 |
| CoDATS | 0.73 | 0.90 | 0.46 | 0.20 | 0.96 | 0.94 | 0.41 | 0.36 | 0.59 | 0.95 | 0.63 |
| RAINCOAT | **0.87** | 0.93 | 0.59 | 0.45 | 0.98 | 0.98 | 0.41 | 0.44 | 0.86 | 0.94 | 0.75 |
| **Ours** | 0.86 | 0.93 | **0.74** | **0.52** | 0.97 | **0.98** | **0.62** | **0.65** | **0.89** | 0.89 | **0.80** |

Table 23: Macro-F1 Score on WISDM for unsupervised domain adaptation.

| Method | 2→32 | 4→15 | 7→30 | 12→7 | 12→19 | 18→20 | 20→30 | 21→31 | 25→29 | 26→2 | Avg |
|---|---|---|---|---|---|---|---|---|---|---|---|
| Source-only | 0.68 | 0.52 | 0.77 | 0.53 | 0.36 | 0.81 | 0.56 | 0.10 | 0.15 | 0.69 | 0.52 |
| CDAN | 0.72 | 0.44 | 0.70 | 0.50 | 0.31 | 0.87 | 0.64 | 0.31 | 0.23 | 0.71 | 0.54 |
| DeepCoral | 0.71 | 0.42 | 0.85 | 0.67 | 0.35 | 0.63 | 0.67 | 0.27 | 0.25 | 0.64 | 0.52 |
| AdaMatch | 0.59 | 0.54 | 0.76 | 0.67 | 0.38 | 0.66 | 0.54 | 0.16 | 0.24 | 0.74 | 0.54 |
| HoMM | 0.63 | 0.42 | 0.62 | 0.55 | 0.39 | 0.63 | 0.60 | 0.30 | 0.26 | 0.54 | 0.49 |
| DIRT-T | 0.65 | 0.41 | 0.78 | 0.56 | 0.39 | 0.67 | 0.65 | 0.28 | 0.21 | 0.54 | 0.54 |
| CLUDA | 0.64 | 0.61 | 0.81 | 0.59 | 0.41 | 0.70 | 0.70 | 0.27 | 0.26 | 0.75 | 0.57 |
| AdvSKM | 0.61 | 0.55 | 0.84 | 0.53 | 0.35 | 0.71 | 0.61 | 0.28 | 0.28 | 0.55 | 0.55 |
| CoDATS | 0.66 | 0.41 | 0.75 | 0.62 | 0.37 | 0.76 | 0.72 | 0.30 | 0.30 | 0.70 | 0.56 |
| RAINCOAT | 0.68 | **0.98** | 0.86 | 0.72 | **0.78** | **0.92** | 0.87 | **0.43** | **0.44** | 0.75 | 0.74 |
| **Ours** | **0.81** | 0.65 | **0.99** | **1.00** | 0.63 | 0.76 | **0.87** | 00.36 | 0.28 | **1.00** | **0.74** |

Table 24: Macro-F1 Score on HHAR-D for unsupervised domain adaptation.

| Method | 0→1 | 0→2 | 0→3 | 0→4 | 1→0 | 1→3 | 1→4 | 2→1 | 3→4 | 4→1 | Avg |
|---|---|---|---|---|---|---|---|---|---|---|---|
| Source-only | 0.61 | 0.27 | 0.25 | 0.33 | 0.44 | 0.43 | 0.46 | 0.32 | 0.85 | 0.38 | 0.43 |
| CDAN | 0.67 | 0.42 | 0.35 | 0.42 | 0.66 | 0.57 | 0.50 | 0.44 | 0.88 | 0.44 | 0.53 |
| DeepCoral | 0.65 | 0.34 | 0.33 | 0.40 | 0.48 | 0.53 | 0.53 | 0.39 | 0.86 | 0.41 | 0.49 |
| AdaMatch | 0.69 | 0.36 | 0.36 | 0.41 | 0.60 | 0.49 | 0.56 | 0.41 | 0.86 | 0.36 | 0.51 |
| HoMM | 0.66 | 0.33 | 0.31 | 0.41 | 0.47 | 0.52 | 0.53 | 0.37 | 0.86 | 0.42 | 0.49 |
| DIRT-T | 0.66 | 0.38 | 0.40 | 0.44 | 0.52 | 0.60 | 0.53 | 0.39 | 0.93 | **0.49** | 0.53 |
| CLUDA | 0.69 | 0.36 | 0.36 | 0.41 | 0.60 | 0.49 | 0.56 | 0.41 | 0.86 | 0.36 | 0.51 |
| AdvSKM | 0.63 | 0.32 | 0.31 | 0.38 | 0.46 | 0.54 | 0.56 | 0.36 | 0.86 | 0.44 | 0.49 |
| CoDATS | 0.71 | 0.38 | 0.44 | 0.39 | 0.70 | 0.61 | 0.53 | 0.38 | 0.90 | 0.44 | 0.55 |
| RAINCOAT | 0.72 | 0.32 | 0.42 | 0.32 | 0.56 | 0.39 | 0.38 | 0.31 | 0.89 | 0.35 | 0.47 |
| **Ours** | **0.76** | **0.53** | **0.49** | **0.56** | **0.81** | **0.67** | **0.59** | **0.44** | **0.93** | 0.41 | **0.62** |

