# OpenReview forum: "Boosting Transferability and Discriminability for Time Series Domain Adaptation"
_NeurIPS.cc/2024/Conference — NeurIPS 2024 poster_

### Official Review · Reviewer_ggDU · 2024-07-09

**Soundness:** 3
**Presentation:** 2
**Contribution:** 3
**Rating:** 6
**Confidence:** 4

**Summary:**

The paper introduces Adversarial CO-learning Networks (ACON) to improve unsupervised domain adaptation (UDA) for time series classification by enhancing the transferability and discriminability of temporal and frequency features. The proposed approach incorporates multi-period frequency feature learning, temporal-frequency domain mutual learning, and domain adversarial learning in temporal-frequency correlation subspaces. Extensive experiments demonstrate ACON's superior performance over existing methods on various time series datasets.

**Strengths:**

- The introduction of Adversarial CO-learning Networks (ACON) that integrates multi-period frequency feature learning, temporal-frequency domain mutual learning, and domain adversarial learning is a novel and promising direction for enhancing unsupervised domain adaptation (UDA) in time series classification.
- The paper demonstrates extensive experiments across a wide range of time series datasets and five common applications, providing strong empirical evidence of the proposed method's effectiveness.
- The paper effectively highlights the distinct properties of temporal and frequency features, proposing mechanisms to leverage their respective strengths for improved transfer learning.

**Weaknesses:**

- Methodological Complexity: The proposed ACON framework is highly complex, involving multiple components and phases (multi-period frequency feature learning, temporal-frequency domain mutual learning, and domain adversarial learning). This complexity may hinder reproducibility and practical implementation, especially in real-world applications.
- Insufficient Scalability Analysis: The paper lacks a detailed discussion on the scalability and computational efficiency of ACON. It remains unclear how the model performs with larger datasets and whether it introduces significant computational challenges.
- Presentation and Clarity Issues: Some sections of the paper are dense and difficult to follow. Simplifying the language and providing clearer explanations would improve readability and accessibility.
- Lack of Detailed Comparative Analysis: While the paper includes experiments against various baselines, it falls short in providing a detailed comparative analysis. More in-depth discussion on why ACON outperforms other methods and the specific contributions of its individual components would strengthen the paper.

**Questions:**

1. The preliminary study assumes that low-frequency components are the most discriminative. However, the choice between phase and amplitude for frequency feature extraction can significantly impact results. Why were only the human activity recognition datasets used for this preliminary study, and how might this assumption affect other types of time series data?
2. The ablation study indicates that the domain loss contributes more significantly to the model's performance than the two KL losses. Could you explain why this is the case, and provide more insights into the individual contributions of each loss component?
3. The paper proposes domain adversarial learning to align features in both source and target domains. However, this seems to conflict with the novel aspect of different alignment strategies based on domain-specific characteristics. Could you clarify how these two approaches are reconciled in your framework?

~~4. How are classification predictions made during inference? The paper mentions a voting mechanism, but additional details on how the final prediction is derived from the individual components would be helpful.~~

5. In Section 4.1, how is the value of k (the number of top amplitudes) determined for multi-period frequency feature learning? Is it dataset-specific, or is there a general guideline for selecting this parameter?

~~6. Can you provide more details on what the learned latent variables represent? Do they capture the essential features that define each class, and how can this be validated?~~

(The issues in Questions 4 and 6 were errors that occurred while I was switching between different pages during the review process of multiple articles. I sincerely apologize to the authors for this.)

**Limitations:**

The authors acknowledge that while ACON enhances transferability and discriminability for time series domain adaptation, it remains unstable when dealing with time series data that exhibit relatively large variances. Addressing this instability is a critical challenge that needs to be tackled in future research.

---

> ### Author Rebuttal · Authors · 2024-08-07
>
> Thank you for your helpful feedback! Due to limited space, additional results are in **PDF attached to Author Rebuttal** and we summarize the weaknesses and questions below.
> ### W1: Reproducibility and implementation in real
> - Compared with existing works, ACON has the widest evaluation scope (8 datasets and 5 common applications) and the sota performance, exhibiting greater potential in solving real problems.
> - Existing UDA methods for time series usually need careful hyperparameter tuning, while ACON fixes all hyperparameters in total loss, making the implementation easier.
> - Code was provided to ensure reproducibility.
> ### W2: Scalability and computational challenges
> - Larger datasets
>   - Comparisons of evaluation scope are in **Table 4 of PDF**. ACON expands the evaluation scope to **more common tasks and more datasets** (8 datasets and 5 common applications) with **longer lengths and more channels**. Performance of ACON on larger and more complex datasets demonstrates its scalability.
> - Computational challenges:
>   - Comparisons of training time are in **Table 5 of PDF**. ACON performs the best, with the second fastest speed, slightly slower than the fastest method. Thus ACON does not introduce additional computational challenges.
> ### W3: Presentation
> We will refine the expression. If the reviewer could point out the specific sections, we would be more grateful and make more targeted improvements.
> ### W4: More in-depth discussion on ACON and other methods
> Existing works are in two categories: general UDA and UDA for time series.
> - General UDA methods ignore the special properties of time series, while ACON fully leverages the respective advantages of temporal and frequency features.
> - Most UDA methods for time series ignore frequency domain. RAINCOAT introduces frequency domain into UDA and assumes that temporal domain and frequency domain are independent, treating them equally. As a result, the improvement brought by frequency domain is incremental. Guided by empirical insight, we propose ACON to fully utilize the advantages of temporal domain and frequency domain:
>   - multi-period feature learning to enhance the discriminability of frequency features
>   - temporal-frequency domain mutual learning to leverage the respective advantages
>   - domain adversarial learning in temporal-frequency correlation subspace to align the source and target distribution
> ### Q1: Only HAR datasets are used in preliminary study
> We apologize for any confusion that may lead the reviewer to think that only HAR datasets are used. We conduct a comprehensive preliminary study on **5** datasets. The results are presented in **Table 1-2 of PDF**. Among them, (1) Results of CAP dataset (SSC task) are in Section 3 of the paper. (2) Results of UCIHAR, HHAR-P and WISDM datasets (HAR task) are in Appendix C.1. (3) We conduct additional experiments on MFD dataset (FD task) in rebuttal to justify the motivation further.
>
> To analyze the impact of phase and amplitude, results of frequency classification experiments using three strategies are in **Table 6 of PDF**.
>
> From Table 1, 2 and 6,  **general** phenomena are observed:
> - Frequency features have better discriminability.
> - Temporal model is more skilled at learning invariant features with domain alignment.
> - Phase can not provide strong discriminative information.
>
> Phenomena of diverse scenarios consistently support our assumption.
> ### Q2: KL losses's improvements are smaller than domain loss
> Domain loss is the key to improving transferability. Without domain loss, transferability of temporal features is weak and cannot significantly enhance transferability of frequency features using KL losses.
> ### Q3: Adversarial learning seems to conflict with mutual learning
> Different mutual learning strategies in source and target domain do not conflict with adversarial learning between source and target. The reasons are:
> - Mutual learning is an alignment (1) between temporal and frequency space (2) at the **sample** level.
>
>   Adversarial learning is an alignment (1) between source and target domain (2) at the **distribution** level.
> - The alignment direction of mutual learning (temporal <-> frequency) is orthogonal to that of adversarial learning (source <-> target). Therefore, mutual learning is not contradictory to adversarial learning in reducing the divergence between source and target.
> - Mutual learning enhances the transferability of frequency features, assisting the alignment of adversarial learning in frequency space.
> - Mutual learning enhances the discriminability of temporal features, enhancing the discriminability of the invariant features learned through adversarial training.
>
> In all, mutual learning and adversarial training share the same goal — to learn domain invariant and discriminative features. To avoid potential confusion caused by the repetition of "alignment" both in mutual learning and adversarial learning, we will change "alignment" in mutual learning to "match".
> ### Q4: A voting mechanism
> The "voting mechanism" did not appear in our paper. We would be grateful if the reviewer could provide more information. To make classification predictions, we apply softmax to the output of the classifier and take the class with the highest score as the final prediction.
> ### Q5: How is the value of k determined?
> In ACON, the original time series length is set as a default period to capture the global information. Upon this, according to the amplitude distribution after min-max normalization, we choose the one with the amplitude ratio > 0.9 as the dominant frequencies. We will add explanations in the next version.
> ### Q6: Latent variables
> The "latent variables" did not appear in our paper. To our knowledge, latent variables refer to the learned representation in Bayesian Generative Models, which is irrelevant to our paper. We would be grateful if the reviewer could provide more information.
>
> We hope our responses were able to address any remaining concerns.

---

> > ### Comment · Reviewer_ggDU · 2024-08-09
> >
> > Thank you for addressing my previous concerns. I appreciate the clarifications and would like to delve deeper into a few points:
> >
> > 1. The observation of asynchronous temporal/frequency features between the source and target domains is quite intriguing. Your experiments confirm this phenomenon, which I find to be a valuable finding.
> > 2. Regarding the KL loss and domain loss, I realize my previous query may not have been clear. From the ablation study, it appears that the domain loss contributes significantly more to the overall performance improvement than the KL loss. The KL loss seems to be an adjustment to align with the authors' previous observations of different domain characteristics. My question is: can domain adaptation tasks be effectively completed with domain loss alone, without relying on the KL loss?
> > 3. Following up on the previous point, I noticed in your rebuttal that you described the relationship between mutual learning and adversarial learning as "orthogonal." While I agree that these two tasks have different objectives, the term "orthogonal" suggests a strong distinction. Considering that these tasks are closely related, is there any theoretical support for describing them as orthogonal?
> >
> > Lastly, I sincerely apologize for the confusion regarding Q4/6. This error was due to my handling multiple paper reviews simultaneously and inadvertently switching pages. I deeply regret any inconvenience this may have caused.

---

> ### Author Response · Authors · 2024-08-09
> **Response for a deeper discussion (Part 1)**
>
> Thank you very much for your quick feedback. We would be delighted to engage in a deeper discussion with you.
>
> ### **Feedback1: The observation of asynchronous temporal/frequency features between the source and target domains is a valuable finding**.
>
> Thank you very much for recognizing our empirical insight as a valuable finding. As a new finding, we believe it can provide insights to guide the design of more algorithms for future works.
>
>
>
> ### **Feedback2: Can domain adaptation tasks be effectively completed with domain loss alone, without relying on the KL loss?**
>
> We will answer this question in 3 subquestions.
>
> ### Subquestion 1. "How to complete domain adaptation tasks effectively?"
>
> To ensure that the reviewer fully understands our discussion, we give some detailed background knowledge in UDA here.
>
> In domain adaptation theory [1], the generalization error $\epsilon_t$ of the target domain can be bounded by:
>
> $\epsilon_t \leq \epsilon_s + d_{\mathcal H\Delta\mathcal H}(P,Q)+\lambda \quad (1)$
>
> $\epsilon_s$ is the source error, $d_{\mathcal H\Delta\mathcal H}(P,Q)$ measures the divergence between the source and target feature distributions, and $\lambda$ is the error of an ideal joint hypothesis $h^*$ defined as $h^∗ = arg \min_{h\in \mathcal H }\epsilon_s(h) + \epsilon_t(h)$, such that:
>
> $\lambda = \epsilon_s(h^*) + \epsilon_t(h^*) \quad (2)$
>
> From Equation (1), we find that if we want to complete domain adaptation tasks effectively, we need to have three small error terms: $\epsilon_s$, $d_{\mathcal H\Delta\mathcal H}(P,Q)$ and $\lambda$. Among them, $\epsilon_s$ is the source error, considering that the model is supervised by source labeled data, $\epsilon_s$ is very small. Thus the key to achieving domain adaptation is to minimize $d_{\mathcal H\Delta\mathcal H}(P,Q)$ and $\lambda$.
>
> $d_{\mathcal H\Delta\mathcal H}(P,Q)$ measures the divergence between the source and target feature distributions, if the features have strong transferability, the divergence between features of the source domain and target domain should be small. Thus, to minimize $d_{\mathcal H\Delta\mathcal H}(P,Q)$, boosting the transferability of features is a promising solution.
>
> In Equation (2), $\lambda$ is the error of an ideal joint hypothesis. It means that given source features and target features, $\lambda$ is the error of the optimal classifier (e.g. linear classifier) to classify features into each class of 2 domains. Thus $\lambda$ can measure the discriminability of features. To minimize $\lambda$, boosting the discriminability of features is a good choice.
>
> With the above background knowledge, the answer to this subquestion is: to complete domain adaptation tasks effectively, we need to minimize $d_{\mathcal H\Delta\mathcal H}(P,Q)$ and $\lambda$​, i.e. **boost transferability and discriminability of features**.
>
>
>
>
>
>
>
> ### Subquestion 2. "How does domain loss help domain adaptation?"
>
> In our original rebuttal to the reviewer ggDU, we answer "Domain loss is the key to improving transferability" but we do not explain it in detail due to the limited space. Here we explain this answer **intuitively and theoretically**.
>
> - Intuitively, adversarial learning is a two-player game between the domain discriminator and the feature extractor. The domain discriminator is trained to distinguish source features from target features and the feature extractor is trained simultaneously to confuse the discriminator. When the game reaches equilibrium, the domain discriminator can no longer distinguish the source features from the target features. At this point, the extracted features are invariant to the change of domains, and the transferability of features is enhanced.
>
> - Theoretically, according to [1], the $\mathcal{H}\Delta \mathcal{H}$-Divergence between the source feature distribution $P$ and target feature distribution $Q$ can be estimated by training the domain discriminator $D$:
>
>   $L_D= \max\limits_{D}\mathbb E_{f\sim P}[D(f)=0]+\mathbb E_{f\sim Q}[D(f)=1] \quad (3)$
>
>   The objective of the feature extractor is to minimize the source error as well as the $\mathcal{H}\Delta \mathcal{H}$-Divergence bounded by Equation (3), boosting the transferability of features.
>
> Overall, the answer to subquestion 2 is: **domain loss is the key to improving transferability of features**.

---

> ### Author Response · Authors · 2024-08-09
> **Response for a deeper discussion (Part 2)**
>
> ### **Feedback2: Can domain adaptation tasks be effectively completed with domain loss alone, without relying on the KL loss?**
>
> ### Subquestion 3. "Can domain loss solve domain adaptation tasks effectively?"
>
> In subquestion 1, we know that to complete domain adaptation tasks effectively, boosting the transferability and discriminability of features is a must.
>
> In subquestion 2, we know that domain loss is the key to improving the transferability of features.
>
> If domain loss can solve domain adaptation tasks effectively, it means domain loss can enhance the discriminability of features. However, the training objective of domain loss is not related to the discriminability of features. Thus there is no guarantee that discriminability of features can be boosted only using domain loss.
>
> Ref[2] reveals that adversarial learning can potentially deteriorate the discriminability since it distorts the original feature distributions when learning domain invariant features.
>
> Thus the answer to the subquestion 3 is: **only using domain loss cannot solve domain adaptation tasks effectively because domain loss may potentially deteriorate the discriminability of features, techniques that can boost the discriminability of features should be introduced.**
>
>
>
>
>
> ### **Feedback3: Is there any theoretical support for describing mutual learning and adversarial learning as orthogonal?**
>
> In our rebuttal,  we state that **the alignment directions** (these words cannot be ignored) of two modules are orthogonal. Intuitively, the alignment direction of mutual learning (temporal <-> frequency) and the alignment direction of adversarial learning (source <-> target) are totally different and easy to understand.
>
> It is quite difficult to prove the orthogonality of two learning modules theoretically, but we can prove it empirically:
>
> Mutual learning and adversarial learning can be treated as different tasks. Following Ref [3], we analyze the individual gradients produced by adversarial learning and mutual learning in the training process. The angles between gradient vectors in mutual learning and gradient vectors in adversarial learning are calculated as in Ref [3], results are in the Table below:
>
> |       | UCIHAR         | HHAR           | WISDM          | Average |
> | ----- | -------------- | -------------- | -------------- | ------- |
> | Angle | $86.16\degree$ | $84.37\degree$ | $88.22\degree$ | $86.25$ |
>
> If the angle between two gradients are close to $90\degree$, these two tasks can be treated as orthogonal. From the above table we can find that in 3 datasets, the average angles between gradient vector in mutual learning and gradient vector are very close to  $90\degree$.
>
> [1] Ben-David et al. "A theory of learning from different domains."  *Machine learning*, 2010.
>
> [2] Liu et al. "Transferable Adversarial Training: A General Approach to Adapting Deep Classifiers." *ICML*, 2019.
>
> [3] Yu et al. "Gradient Surgery for Multi-Task Learning." *NeurIPS*, 2020.
>
>
>
> Thanks for your valuable comments. We hope our responses were able to address any remaining concerns. Please do let us know if you have any further questions as well as what would be expected for score improvement.

---

> ### Comment · Reviewer_ggDU · 2024-08-10
>
> Thank you for your response. I have reviewed the content, and these replies effectively address my current concerns. I believe the authors have a solid understanding and insight into their research work. Additionally, they offer a fresh perspective in the field of UDA, making this a commendable contribution.
>
> I've edited the comments and the score.

---

> > ### Author Response · Authors · 2024-08-10
> > **Response to Reviewer ggDU**
> >
> > Thank you for the timely response and raising the score to 6. We are pleased to know that we have successfully addressed your concerns. We sincerely appreciate your recognition of our work as a commendable contribution that offers a fresh perspective in the field of UDA.
> >
> > We will revise our paper according to your feedback and the results in the rebuttal, including adding more details about how to determine top-k periods, providing more comparisons about the evaluation scope and training speed, and discussing the enhancement between different modules.
> >
> > Thank you again for your invaluable feedback and dedicated time to review our paper. If you have any questions, please send them to us, we look forward to discussing with you to further improve our work.

---

### Official Review · Reviewer_U97c · 2024-07-12

**Soundness:** 3
**Presentation:** 3
**Contribution:** 3
**Rating:** 6
**Confidence:** 4

**Summary:**

This paper studies the Unsupervised Domain Adaptation (UDA) problem in time series classification. The authors first proposed an insight that temporal features enhance transferability while frequency features enhance discriminability. Based on the insight, the authors designed a model that leverages temporal and frequency features. Other techniques include but not limited to Knowledge Distillation and Domain Adversarial Learning. The evaluations show SOTA performance, and the ablation study indicates that the designed components are helpful.

**Strengths:**

1.	Empirical insights into temporal and frequency features of time series data in terms of transferability and discriminability.
2.	It is interesting to see how the authors managed to achieve better transferability and discriminability through knowledge distillation.
3.	The designed temporal-frequency subspace for domain adversarial learning showed notable performance increase.

**Weaknesses:**

1.	This paper is derived from the empirical insight of temporal and frequency features. It would be better if the authors can explain a little bit about why temporal or frequency features excel in different properties.
2.	The preliminary experiment on transferability and discriminability could use more datasets and more analysis to justify the motivation of the paper.
3.	Since there are other papers utilizing temporal and frequency features as mentioned in the related work, it would be better if the author can illustrate the differences and the advantages of the current design compared to other works.
4.	It would be helpful to improve clarity if the author can explain the “index” in figure 1.
5.	During the illustration of the model proposed, the author did not cover the design of the domain discriminator.

**Questions:**

It’s a great idea to exploit the advantages of both temporal and frequency features using distillation. But I also wonder why a simple fusion of temporal and frequency features is not used and compared. Will the fused features maintain both good transferability and discriminability?

**Limitations:**

Yes

---

> ### Author Rebuttal · Authors · 2024-08-07
>
> Thank you very much for your valuable feedback! The additional results are presented in **the PDF attached to Author Rebuttal** and we summarize the original weaknesses or questions.
>
> ### W1: The intuition behind empirical insight
>
> Here we provide an intuition from the perspective of energy.
>
> According to Parseval-Plancherel identity [1], the energy of a time series in the temporal domain is equal to the energy of its representation in the frequency domain. By FFT, the energy of raw temporal signals in Euclidean space is transformed to the frequency data in Hilbert space. In Euclidean space, the energy is allocated to every time step, while in Hilbert space, the energy is mainly allocated to several dominant frequencies.
>
> This means that in the frequency domain, the model needs to focus its attention on a local region to capture the dominant frequencies, while in the temporal domain, the model needs to distribute its attention globally to extract discriminative patterns, which makes the temporal modeling still challenging. In contrast, in the frequency domain,  the discriminative patterns have been "**pre-extracted**" via FFT, which makes the frequency model more easily extract discriminative features.
>
> When a shift happens, the frequency model with local attention may encounter **a dramatical local shift** (e.g. a shift in the dominant frequencies of a certain class), leading to worse transferability. In contrast, the temporal features with a more evenly distributed energy are more resistant to the shift and exhibit better transferability.
>
> [1] Plancherel, et al. "Contribution à ľétude de la représentation d’une fonction arbitraire par des intégrales définies." 1910.
>
> ### W2: More datasets for preliminary study
>
> We conduct a comprehensive preliminary study on a wide range of datasets (**5 datasets from 3 tasks**). The results are comprehensively presented in **Table 1-2 of PDF**. Among them, (1) Results of SSC task (CAP dataset) are included in Section 3 of the paper. (2) Results of HAR task (UCIHAR, HHAR-P and WISDM datasets) are included in Appendix C.1. (3) We conduct additional experiments on FD task (MFD dataset) in rebuttal to further justify the motivation.
>
> From Table 1 and 2, we can observe **general** experimental phenomena:
>
> - Frequency features have better discriminability.
> - Temporal model has a better ability to learn invariant features in the progress of alignment.
>
> The phenomena of different types of data consistently support our empirical insights, demonstrating its generalizability.
>
> ### W3: Comparison with other works
>
> To our knowledge, RAINCOAT is currently the only work that introduces the frequency domain into time series domain adaptation.
>
> For the extraction of frequency features, RAINCOAT treats amplitudes and phases equally, while ACON discards the phase and proposes multi-period frequency feature learning to enhance discriminability.
>
> For the utilization of temporal and frequency features, RAINCOAT assumes them are independent and treats them equally, resulting in the improvement brought by the frequency domain being incremental. In contrast, based on the respective advantages, ACON proposes temporal-frequency domain mutual learning and co-alignment in temporal-frequency correlation subspace, achieving a performance boost.
>
> ### W4 & W5: Index & Design of domain discriminator
>
> Thank you very much for pointing out the confusion and the missing.
>
> For Figure 1(a), the index refers to time steps or frequencies. For Figure 1(b), the index refers to different domains contained in the CAP dataset. For Figure 1(c), the index refers to different source-target pairs selected from the CAP dataset.
>
> We adopt a 3-layer linear as the domain discriminator and ReLU as the activation function, which is consistent with existing UDA works. We will add the explanations in the later version.
>
> ### Q1: Comparision with a simple fusion
>
> We conduct exploratory experiments to verify whether a simple fusion in classification or alignment module can produce better performance. The results are presented in Table 3 of the PDF. Due to limited space, a more detailed explanation is included in **the caption of Table 3**.
>
> In classification module, by comparing the 4th, 5th and 6th rows in Table 3, we can observe that concatenation features do not achieve better performance. Intuitively, when we use the concatenation feature for classification, the final prediction is **a simple average** of the temporal prediction and the frequency prediction. Ideally, the extracted features can only maintain suboptimal discriminability and suboptimal transferability. In contrast, our mutual learning allows the frequency domain and temporal domain to become teachers with different advantages, and transfer knowledge to each other. In this way, each other's secondary quantities (e.g. the estimates of the probabilities of the next most likely classes) are transferred [1], enhancing the discriminability and transferability.
>
> In alignment module, by comparing the 3rd and 6th rows, we can observe that with the concatenation DANN significantly underperforms our method. Intuitively, when we use the concatenation feature for alignment, $\mathbf f_F$ with the worse transferability provides $g_D$ with rich domain-label relevant information. In this case, $g_D$ only needs to focus on the game with $\psi_{F}$ in the frequency domain, ignoring the domain adversarial learning in the temporal domain. Based on the experimental performance and intuition, we discard the simple fusion and opt for co-alignment in the temporal-frequency correlation subspace.
>
> [1] Hinton, et al. "Distilling the knowledge in a neural network." *Deep Learning Workshop in NeurIPS*, 2014.
>
> We hope our responses were able to address any remaining concerns. Please do let us know if you have any further questions as well as what would be expected for score improvement.

---

> > ### Comment · Reviewer_U97c · 2024-08-12
> >
> > Thanks for the response. I am leaning to accept this paper and would like to maintain my positive score.

---

> > > ### Author Response · Authors · 2024-08-12
> > > **Response to Reviewer U97c**
> > >
> > > Thank you for the timely response and efforts in reviewing our paper. We sincerely appreciate your recommendation to accept our paper.
> > >
> > > We will revise our paper according to your feedback and the results in the rebuttal, including adding:
> > >
> > > - The intuition behind the empirical insight.
> > > - More datasets for preliminary study.
> > > - More details: a more detailed comparison with other works, the design of the domain discriminator, and the explanation of the "index".
> > > - A comprehensive comparison of simple fusion.
> > >
> > > Thank you again for your invaluable feedback and dedicated time to review our paper. If you have any questions, please send them to us, we look forward to discussing with you to further improve our work.

---

### Official Review · Reviewer_5Lht · 2024-07-13

**Soundness:** 3
**Presentation:** 3
**Contribution:** 3
**Rating:** 6
**Confidence:** 4

**Summary:**

The paper shows that temporal features and frequency features should not be equally treated in model training, the expression of those features are different. Frequency features show strong discriminability and temporal features show strong transferability. Then the author propose the Multi-period frequency feature learning; Temporal-frequency domain mutual learning and Domain adversarial learning in temporal-frequency correlation subspace to leverage the advantages. The proposed results are impressively good.

**Strengths:**

The proposed differences of frequency feature and temporal feature is interesting.
The proposed methods are novel.
The results are impressively good.

**Weaknesses:**

The motivation of the proposed methods should be clarified.

1. Why the alignment between the temporal predictions and the frequency predictions can leverage the respective advantages of the temporal domain and frequency domain? There is no guarantee to keep both advantages.
2. Why the Multi-period frequency feature learning can enhancing the discriminability of the frequency domain?
3. Why Domain adversarial learning can let the model learn transferable representations?

**Questions:**

See weaknesses.

**Limitations:**

The authors have addressed the limitations.

---

> ### Author Rebuttal · Authors · 2024-08-07
>
> Thank you for your valuable feedback! We are grateful that the reviewer highlights our method's novelty and solid experiments. We would be happy to explain the motivations behind the different components in more detail.
>
> ### W1: Why the alignment between the temporal predictions and the frequency predictions can leverage the respective advantages of the temporal domain and frequency domain?
>
> The alignment between the temporal predictions and the frequency predictions is inspired by knowledge distillation (KD). KD aligns the student model's predictions to the teacher model's predictions to transfer knowledge from teacher to student. Existing studies have proven that learning to mimic the teacher is easier than learning the target function directly, and the student can match or even outperform the teacher [1].
>
> Given the respective advantages of the frequency domain and temporal domain,  ACON allows the frequency domain and temporal domain to each become a teacher with different advantages. By minimizing the KL divergence with different directions, the respective advantages of the frequency domain and temporal domain are transferred to each other. In this way, the temporal domain and the frequency domain can not only learn the true label but also learn each other's secondary quantities (e.g. their estimates of the probabilities of the next most likely classes) [2], enhancing the discriminability and transferability.
>
> [1] Romero, Adriana, et al. "Fitnets: Hints for thin deep nets." *ICLR*, 2015.
>
> [2] Hinton, et al. "Distilling the knowledge in a neural network." *Deep Learning Workshop in NeurIPS*, 2014.
>
> ### W2: Why the Multi-period frequency feature learning can enhance the discriminability of the frequency domain?
>
> The real-world time series usually presents multi-periodicity, which is reflected in the frequency domain as the presence of a few dominant frequencies with significantly larger amplitudes. Data from different periods can have different discriminative patterns. By multi-period frequency feature learning, ACON extracts these discriminative patterns derived from different periods.
>
> ### W3: Why Domain adversarial learning can let the model learn transferable representations?
>
> We provide explanations from two perspectives: intuition and theory.
>
> - Intuitively, adversarial learning is a two-player game between the domain discriminator and the feature extractor. The domain discriminator is trained to distinguish source features from target features and the feature extractor is trained simultaneously to confuse the discriminator. When the game reaches equilibrium, the domain discriminator can no longer distinguish the source features from the target features. At this point, the extracted features are invariant to the change of domains.
>
> - Theoretically, according to [1], the $\mathcal{H}\Delta \mathcal{H}$-Divergence between the source feature distribution $P$ and target feature distribution $Q$ can be estimated by training the domain discriminator $D$:
>
>    $L_D= \max\limits_{D}\mathbb E_{f\sim P}[D(f)=0]+\mathbb E _{f\sim Q}[D(f)=1]$    （1）
>
>
>
>   The objective of the feature extractor is to minimize the source error as well as the $\mathcal{H}\Delta \mathcal{H}$-Divergence bounded by Equation (1), resulting in transferable representations.
>
>   [1] Ben-David, et al. "A theory of learning from different domains."  *Machine learning*, 2010.
>
> Thank you again for the valuable feedback! We hope our responses were able to address any remaining concerns. Please do let us know if you have any further questions as well as what would be expected for score improvement.

---

> ### Author Response · Authors · 2024-08-12
> **Looking forward to your feedback**
>
> Dear Reviewer 5Lht,
>
> Thank you for your valuable comments.
>
> We have made an extensive effort trying to address your concerns. In our response:
>
> - From the perspective of Knowledge Distillation, we clarify why the alignment between the temporal predictions and the frequency predictions can leverage the respective advantages.
> - Based on the inherent characteristic of time series, we explain why the Multi-period frequency feature learning can enhance the discriminability of the frequency domain.
> - From the intuitive perspective and the theoretical perspective, we provide explanations of why Domain adversarial learning can let the model learn transferable representations.
>
> We hope our response can address your concerns. If you have any further concerns or questions, please do not hesitate to let us know, and we will be more than happy to address them promptly.
>
> All the best,
>
> Authors

---

> ### Comment · Reviewer_5Lht · 2024-08-12
>
> Thanks for the explanations, I will keep my rating to 6.

---

> > ### Author Response · Authors · 2024-08-12
> > **Response to Reviewer 5Lht**
> >
> > Thank you for your invaluable feedback and dedicated time to review our paper.
> >
> > We will revise our paper according to your feedback, adding more explanation on why each component works. If you have any further questions, please send them to us, we look forward to discussing with you to further improve our work.

---

### Official Review · Reviewer_uayq · 2024-07-15

**Soundness:** 3
**Presentation:** 3
**Contribution:** 3
**Rating:** 7
**Confidence:** 3

**Summary:**

This paper uncovers an empirical insight in time series domain adaptation - frequency features are more discriminative within a specific domain, while temporal features show better transferability across domains. Based on this insight, it develops ACON (Adversarial CO-learning Networks), which achieves clear improvement over baselines.

**Strengths:**

- Empirical insight: it is great to see such empirical analysis to provide an interesting insight about the properties of frequency/temporal features.
- Technical solution, which leverages the insight to improve accuracy.
- Experiments: it is good to see solid experimental efforts in the paper and appendix.

**Weaknesses:**

- It would be better to explain (or explore) “why” behind the insight? Is it because the distinguished frequency patterns shift across domains dramatically?

- Usually, neural networks can capture some frequency patterns from time series data. Why it avoids most distinguished frequency patterns, but extracts more transferable ones?

**Questions:**

See weakness.

**Limitations:**

The limitation of stability has been discussed.

---

> ### Author Rebuttal · Authors · 2024-08-07
>
> Thank you for your helpful feedback! We are grateful that the reviewer highlights our empirical insight, technical solution and solid experiments. We would be delighted to explore the underlying reasons behind the empirical insight with the reviewer.
>
> ### W1: Why the frequency features have worse transferability? Is it because the distinguished frequency patterns shift across domains dramatically?
>
> A1:
>
> Here we provide an intuition from the perspective of energy.
>
> According to Parseval-Plancherel identity [1], the energy of a time series in the temporal domain is equal to the energy of its representation in the frequency domain. By FFT, the energy of raw temporal signals in Euclidean space is transformed to the frequency data in Hilbert space. In Euclidean space, the energy is allocated to every time step, while in Hilbert space, the energy is mainly allocated to several dominant frequencies.
>
> This means that in the frequency domain, the model needs to focus its attention on a local region to capture the dominant frequencies, while in the temporal domain, the model needs to distribute its attention globally to extract discriminative patterns. When a shift happens, the frequency model with local attention may encounter **a dramatical local shift** (e.g. a shift in the dominant frequencies of a certain class), leading to worse transferability. In contrast, the temporal features with a more evenly distributed energy are more resistant to the shift and exhibit better transferability.
>
> ### W2: Why the temporal model avoids to capture most distinguished frequency patterns?
>
> A2:
>
> As mentioned in A1, with a more evenly distributed energy, the temporal model needs to distribute its attention globally to extract discriminative patterns (eg. multi-period or trend). This global extraction ability is often highly correlated with the model architecture and depth, which makes the temporal modeling still challenging. In contrast, in the frequency domain,  the discriminative patterns have been "**pre-extracted**" via FFT, which makes the frequency model more easily extract discriminative features.
>
> Our experimental results also support this point. We compare the sota models of temporal modeling (TimesNet [2] and DLinear [3] ) with 1D-CNN on source domain classification (results are presented in the following table), and it was surprisingly found that these more complex models do not show a significant improvement in UDA settings. This indicates that the existing temporal models still do not have an ideal ability to extract the most discriminative patterns.
>
> Based on intuition and empirical insight, we believe the mutual enhancement between the temporal domain and the frequency domain can be a promising solution.
>
> | Temporal model | UCIHAR | HHAR-P | WISDM |
> | -------------- | ------ | ------ | ----- |
> | 1D-CNN         | 86.18  | 97.01  | 95.63 |
> | TimesNet       | 89.85  | 95.07  | 88.42 |
> | DLinear        | 67.18  | 67.03  | 75.30 |
>
> [1] Plancherel, Michel, and Mittag Leffler. "Contribution à ľétude de la représentation d’une fonction arbitraire par des intégrales définies." *Rendiconti del Circolo Matematico di Palermo*, 1910.
>
> [2] Wu, Haixu, et al. "TimesNet: Temporal 2D-Variation Modeling for General Time Series Analysis." *ICLR*, 2023.
>
> [3] Zeng, Ailing, et al. "Are transformers effective for time series forecasting?" *AAAI*, 2023.
>
> Thank you again for providing insightful comments which helped us to improve our paper, and we hope our responses were able to address any remaining concerns. Please do let us know if you have any further questions as well as what would be expected for score improvement.

---

> ### Author Response · Authors · 2024-08-12
> **Looking forward to your feedback**
>
> Dear Reviewer uayq,
>
> Thank you for your valuable comments.
>
> We have made an extensive effort trying to address your concerns. In our response:
>
> - From the perspective of energy, we provide an intuition behind the empirical insight.
> - We further demonstrate that the mutual enhancement between the temporal domain and the frequency domain is a promising solution by comparing the sota models of temporal modeling with 1D-CNN.
>
> We hope our response can address your concerns. If you have any further concerns or questions, please do not hesitate to let us know, and we will be more than happy to address them promptly.
>
> All the best,
>
> Authors

---

### Author Rebuttal · Authors · 2024-08-07

Thank you to all reviewers for the thoughtful feedback.

We are pleased that all four reviewers agree with our empirical insight, method novelty and solid experiments. We are also delighted that reviewers recognized this study as a promising solution to UDA for time series.

In response to reviewers' comments, we perform 4 additional experiments and organize the results into 6 new tables. Due to the limited space, the new tables are included in **the PDF attached to this Author Rebuttal**. We hope these updates address all key concerns and clarify the significance of our work.

We respond to all your comments below in the individual replies, where we directly address comments raised by individual reviewers. Please note that in our response, all experimental results are presented in the PDF attached to this Author Rebuttal.  In addition, we carefully proofread the paper and edit the paper for clarity. We will include further details in the next version (new revisions during the rebuttal are not allowed).

Thank you again for your thoughtful feedback. If you have any further concerns or questions, please do not hesitate to let us know.

**Table 1-2**: More datasets for preliminary study (Reviewer U97c and Reviewer ggDU)

**Table 3**: Comparison with a simple fusion  (Reviewer U97c)

**Table 4**: Comparison of evaluation scope (Reviewer ggDU)

**Table 5**: Comparison of accuracy and speed (Reviewer ggDU)

**Table 6**: Three different strategies in the frequency domain (Reviewer ggDU)

---

### Decision · Program_Chairs · 2024-09-25

**Decision:**

Accept (poster)

**Comment:**

Reviewers agree on acceptance based on meaningful empirical insights, effective technical solutions, and notable performance. The authors are encouraged to include discussions and results in the rebuttal to the final version.